# Alterations of gut microbiota contribute to the progression of unruptured intracranial aneurysms

Hao Li [1,8], Haochen Xu[1,8], Youxiang Li[2,8], Yuhua Jiang[2,8], Yamin Hu[3], Tingting Liu[1], Xueqing Tian[1], Xihai Zhao [4], Yandong Zhu [4], Shuxia Wang[5], Chunrui Zhang[6], Jing Ge[1], Xuliang Wang[1], Hongyan Wen[1], Congxia Bai[1], Yingying Sun[1], Li Song[1], Yinhui Zhang[1], Rutai Hui[1], Jun Cai[7] & Jingzhou Chen [1✉]

Unruptured intracranial aneurysm (UIA) is a life-threatening cerebrovascular condition. Whether changes in gut microbial composition participate in the development of UIAs remains largely unknown. We perform a case-control metagenome-wide association study in two cohorts of Chinese UIA patients and control individuals and mice that receive fecal transplants from human donors. After fecal transplantation, the UIA microbiota is sufficient to induce UIAs in mice. We identify UIA-associated gut microbial species link to changes in circulating taurine. Specifically, the abundance of *Hungatella hathewayi* is markedly decreased and positively correlated with the circulating taurine concentration in both humans and mice. Consistently, gavage with *H. hathewayi* normalizes the taurine levels in serum and protects mice against the formation and rupture of intracranial aneurysms. Taurine supplementation also reverses the progression of intracranial aneurysms. Our findings provide insights into a potential role of *H. hathewayi*-associated taurine depletion as a key factor in the pathogenesis of UIAs.

[1] State Key Laboratory of Cardiovascular Disease, Fuwai Hospital, National Center for Cardiovascular Diseases, Chinese Academy of Medical Sciences and Peking Union Medical College, Beijing 100037, China. [2] Department of Interventional Neuroradiology, Beijing Neurosurgical Institute and Beijing Tiantan Hospital, Capital Medical University, Beijing 100050, China. [3] Department of Cardiology, Cangzhou Central Hospital, Cangzhou 061000, China. [4] Department of Biomedical Engineering, School of Medicine, Tsinghua University, Beijing 100084, China. [5] Chinese PLA General Hospital and Chinese PLA Medical College, Beijing 100853, China. [6] Novogene Bioinformatics Institute, Beijing 100083, China. [7] Hypertension Center, Fuwai Hospital, State Key Laboratory of Cardiovascular Disease of China, National Center for Cardiovascular Diseases of China, Chinese Academy of Medical Sciences and Peking Union Medical College, Beijing 100037, China. [8] These authors contributed equally: Hao Li, Haochen Xu, Youxiang Li, Yuhua Jiang. ✉email: chendragon1976@aliyun.com

Unruptured intracranial aneurysms (UIAs) are being detected more often by cross-sectional imaging techniques and are an important healthcare burden. Findings from an analysis of 68 studies reporting data from 83 study populations and 1450 UIAs in 94912 patients reveal an overall prevalence of intracranial aneurysms of 3.2%[1]. Rupture of an intracranial aneurysm is the underlying cause of 80–85% of non-traumatic subarachnoid hemorrhages[2], often leading to the loss of many years of productive life. However, the precise mechanisms that contribute to the pathogenesis of UIAs remain to be elucidated. Although studies of the genetic predisposition for UIAs have implicated genes such as *CDKN2A, SOX17*, and *ADAMTS15*[3], a large population-based heritability study estimated the heritability of intracranial aneurysms and subarachnoid hemorrhages to be 41%[4], suggesting that environmental factors also contribute.

Accumulating evidence indicates that the gut microbiota is a pivotal environmental factor that influences host metabolism and immune homeostasis. Recently, considerable interest has been focused on the role of the human gut microbiota in cardiovascular diseases such as hypertension[5], heart failure[6], and atherosclerosis[7]. Studies have shown that, in these diseases, the gut microbiota produce numerous metabolites that are absorbed into the systemic circulation, where they are further metabolized by host enzymes, leading to target organ damage[8,9]. A recent study reported that depletion of the gut microbiota by antibiotics reduces the incidence of intracranial aneurysms in mice[10]. Nevertheless, direct evidence for altered gut microbial composition in UIA patients and the precise underlying mechanisms are still lacking.

Metagenome-wide association studies (MWAS) can provide valuable information about the presence of characteristic gut microbiota. In the present study, we sequence stool samples, representative of the gut microbiota, from healthy controls and UIA patients and perform a MWAS. We then perform serum metabolomic analyses of the participants' metabolic profiles and find UIA-associated microbial species and explore their effects on host amino acid and fatty acid metabolism. Using a mouse model of fecal transplantation, we uncover evidence for the potential effects of altered *Hungatella hathewayi* abundance on circulating taurine levels and on the increased occurrence of UIAs.

## Results

**UIA-associated genes and taxonomic changes identified by MWAS.** To investigate the gut microbiota in UIA patients, we performed metagenomic shotgun sequencing on a total of 280 fecal samples (200 samples from 100 UIA patients and 100 controls in the first cohort; 80 from 40 UIA patients and 40 controls in the second cohort, Supplementary Fig. 1). For each sample, a majority of high-quality sequencing reads were assembled de novo into long contigs or scaffolds, which were used for gene prediction, taxonomic classification, and functional annotation (Supplementary Data 1–3).

We first investigated the richness and evenness of the gut microbiota in the first cohort. Rarefaction analysis was used to estimate the total number of genes that could be identified from these samples; this showed that the gene richness approached saturation in each group (Fig. 1a). Neither genus counts nor α-diversity significantly differed between the two groups ($P = 0.248$ for counts, $P = 0.276$ for diversity; Wilcoxon rank-sum test; Fig. 1b, c). Ordination of Bray–Curtis dissimilarity by principal coordinate analysis (PCoA) revealed separation of the two groups (Fig. 1d). This separation was associated with differences in community composition as evaluated by analysis of similarity (ANOSIM) ($R = 0.061$, $P = 0.001$; Fig. 1e), which indicated a less heterogeneous community structure among UIA patients than among controls.

It is noteworthy that 145 genera were differentially enriched in the UIA or control groups (Supplementary Data 4). *Bacteroides, Parabacteroides, Ruminococcus*, and *Blautia* were significantly overrepresented in UIA patients, while *Faecalibacterium, Eubacterium, Collinsella*, and *Lactobacillus* were overrepresented in the controls (Fig. 1f). To further evaluate whether regional factors are potential confounders affecting the features of the gut microbiota, PCoA based on the features of the gut microbiota at the genus level was applied to discriminate control individuals from different areas. As expected, PCoA at the genus level showed a similar spatial distribution between the two groups (Supplementary Fig. 2), indicating that control individuals from different areas shared similar features of gut microbiota.

**Gut microbial species associated with UIAs.** To further identify the microbial species associated with UIAs, MetaPhlAn2 (metagenomic phylogenetic analysis)[11,12] was run to identify the taxonomic abundances at the species level. The α-diversity at the species level did not significantly differ between the two groups ($P = 0.824$; Wilcoxon rank-sum test; Supplementary Fig. 3). In the first cohort, a total of 47 species differed significantly in abundance between the UIA and control samples (Supplementary Data 5), 38 of which were more abundant in the UIA samples. A bacterial community of *Bacteroides* sp., such as *B. thetaiotaomicron, B. massiliensis, B. nordii, B. intestinalis*, and *B. cellulosilyticus*, was significantly enriched in the UIA patients compared with the controls. The abundance of *Odoribacter splanchnicus*, which has been reported to show a positive correlation with the consumption of red meat and a negative correlation with the consumption of fruits and vegetables[13], was higher in UIA patients. Furthermore, we found that the abundance of *Clostridium sp.*, including *C. bartlettii, C. nexile*, and *C. bolteae*, was also significantly enriched in the UIA patients. However, *C. hathewayi*, which was reclassified as *H. hathewayi* in 2014[14], was the only *Clostridium sp.* enriched in the control group. Among the 47 differentially abundant species, 8 (17.0%) were also differentially enriched in UIA patients compared with controls in the second cohort (Fig. 2 and Supplementary Data 6).

Moreover, we clustered the genes that displayed significant differences in abundance between the two groups into metagenomic linkage groups (MLGs). A total of 220 of the MLGs differed significantly in abundance between the UIA and control samples (adjusted $P < 0.05$, Wilcoxon rank-sum test; Supplementary Data 7), 137 of which were more abundant in the UIA samples.

In addition to abundance differences between the UIA and control samples, the MLGs also showed differences in network structure, which was constructed by SparCC (Sparse Correlations for Compositional data) (Supplementary Fig. 4). We found 256 and 171 positive significant correlations in UIA and control-enriched MLGs, respectively (correlation coefficient > 0.5 and Benjamin–Hochberg corrected $P < 0.01$). Five negative significant correlations were also found among MLGs (correlation coefficient < −0.5 and Benjamin–Hochberg corrected $P < 0.01$).

Moreover, we assessed the divergence of gut microbiota composition to explore the correlation of microbial abundance with host factors and found no remarkable regularity of bacterial abundance based on age, BMI, sex, smoking, or alcohol consumption (Supplementary Fig. 5). We further selected 61 of the 220 MLGs to distinguish between individuals with and without UIAs, and the accuracy of cross-validation reached 81%, indicating that these 61 MLGs were distinct UIA-associated features of the gut microbiota (Supplementary Fig. 6, Supplementary Data 8). It is

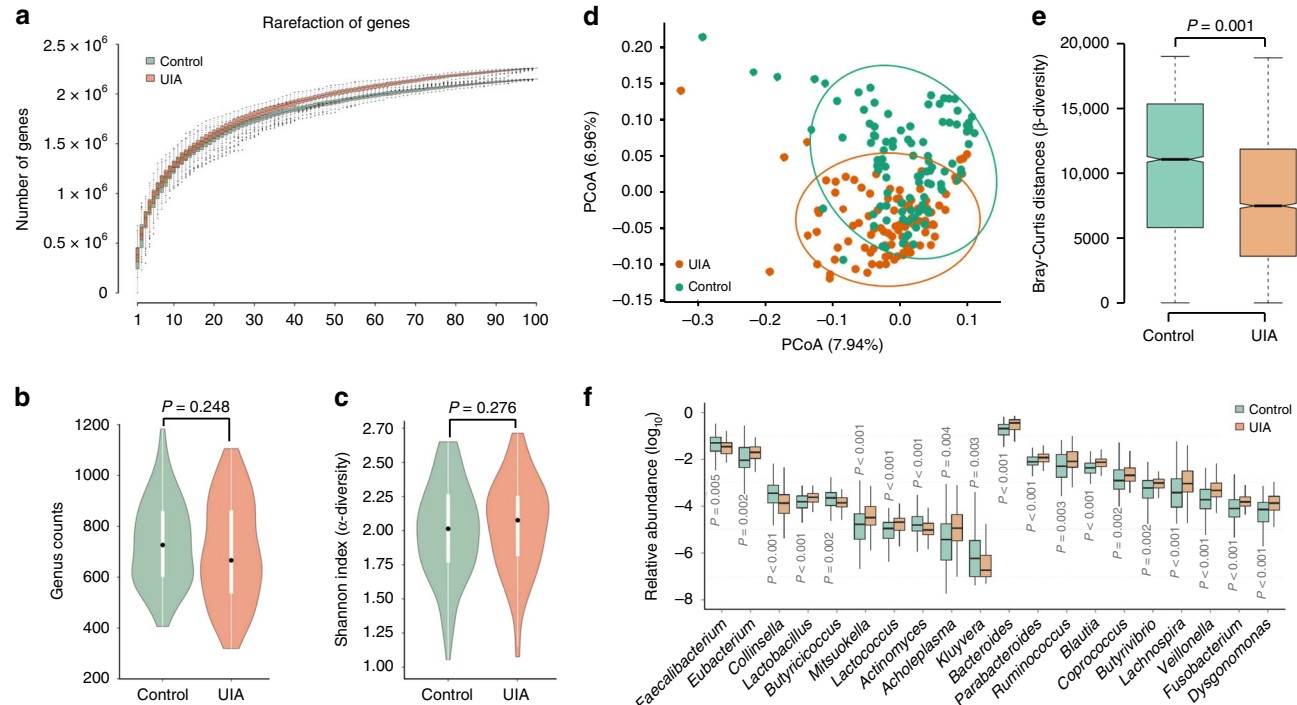

**Fig. 1 Gut microbial alterations in UIA patients. a** Rarefaction curves for gene number in controls ($n = 100$) and UIA patients ($n = 100$) after 100 random samplings. **b, c** Comparison of microbial genus counts and α-diversity (as assessed by the Shannon index) based on the genus profiles in the two groups. Interquartile ranges (IQRs; thick bars), medians (open dots on the bars), the lowest and highest values within 1.5 times IQR from the first and third quartiles (lines above and below the bars). **d** Principal coordinate analysis of samples from UIAs and controls. **e** β-diversity (as assessed by the Bray–Curtis distances) based on the genus profiles in the two groups ($n = 100$). **f** Relative abundances of the most abundant genera that showed significant differences between UIA patients and controls. For (**a**), (**e**), and (**f**), boxes represent the IQRs between the first and third quartiles, and the line inside the box represents the median; whiskers represent the lowest or highest values within 1.5 times IQR from the first or third quartiles. For (**b**), (**c**), and (**f**), the two-tailed Wilcoxon rank-sum test was used. For (**e**), ANOSIM analysis was performed. Source data are provided as a Source Data file.

worth noting that the discrimination between UIAs and control samples for these 61 MLGs was further validated in the second cohort. Seventeen MLGs (27.9%) were also differentially enriched in the UIA patients compared with the controls (Supplementary Fig. 7).

Notably, among the 8 MetaPhlAn2-associated species and the 17 MLG-associated species that differed in abundance between the two cohorts, we found that *H. hathewayi*, *O. splanchnicus* and *Alistipes putredinis* considerably overlapped; these were the most representative microbial changes in the deteriorating gut microbiota of the UIA patients.

**Identification of UIAs based on gut microbiota**. To illustrate the microbial signature and explore the diagnostic value of the composition of the gut microbiota in relation to UIA, we constructed a random forest classifier from the 200 UIA and control samples of the first cohort. Four-fold cross-validation and receiver operating characteristic curves for distinguishing UIA patients from controls were developed. We were able to detect UIA patients accurately based on the 47 species that were differentially abundant (as identified by MetaPhlAn2) in the first cohort, as indicated by an area under the receiver operating curve (AUC) of up to 0.86 and a 95% confidence interval (CI) of 0.81–0.91 (Fig. 3a). Consistent with these results, the classification error remained relatively low in the second cohort (80 UIA and control samples), with an AUC of 0.72 and a 95% CI of 0.65–0.79 (Fig. 3b), indicating that information on the gut microbiota might be used to identify UIA patients. Overall, the presence of UIA-associated features in the gut microbiota offers further evidence

for a dysbiotic gut microbiota and highlights the potential of gut microbiota biomarkers for the noninvasive detection of populations with UIAs.

**Functional characterization of the UIA microbiome**. We next aimed to characterize the species-level functional profiling in UIA through the use of HUMAnN2[15] (the HMP Unified Metabolic Analysis Network 2) pipeline. This allows the assessment of molecular activities of microbial communities at the whole-pathway level, which provides a more efficient and accurate profiling of abundance in microbial pathways from a community based on metagenomic data. We identified 46 metabolic pathways that were differentially abundant between UIA patients and controls (Fig. 4a, Supplementary Data 9). Within these 46 pathways, those contributing to amino acid and fatty acid metabolism were adequately represented. Specifically, the microbiome of UIA patients was found to be significantly dominated by unsaturated fatty acid biosynthesis. Moreover, the biosynthesis of amino acids such as threonine, isoleucine, lysine, and methionine dominated the metabolic landscape of the microbiome in controls. The rest of the pathways were mainly represented by glucose and nucleotide metabolic pathways.

The functional characterization of the gut microbiome was further validated by using the KEGG database based on MLGs. We identified 25 KEGG orthologues that differed in abundance between UIA patients and controls (adjusted $P < 0.05$, Wilcoxon rank-sum test; Supplementary Data 10). Notably, unsaturated fatty acid biosynthesis, amino acid (methionine, tryptophan, pyruvate; and isoleucine) metabolism and glycolysis were overlapping

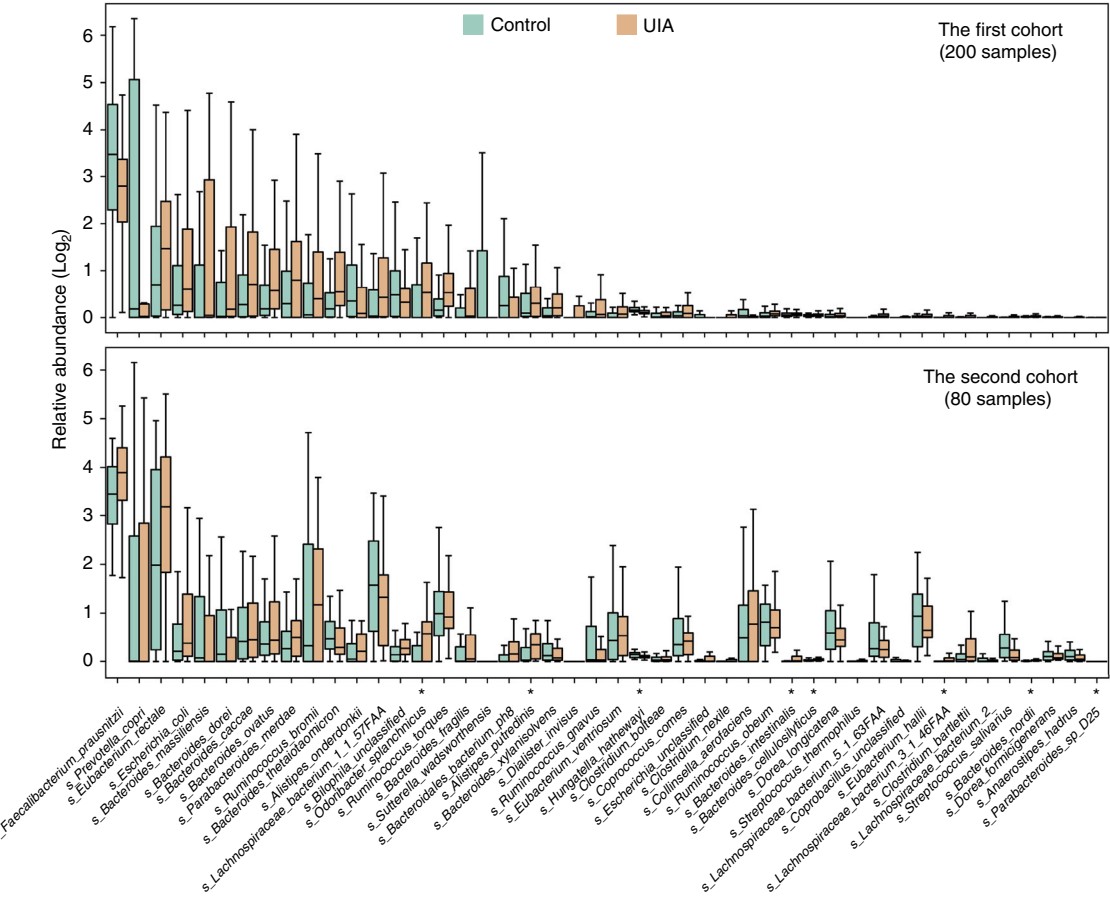

**Fig. 2 Alterations of gut microbial species in UIA patients.** Turquoise and orange, abundance of control- and UIA-enriched species; *$P < 0.05$, for both the first ($n = 100$) and second cohorts ($n = 40$) (two-tailed Wilcoxon rank-sum test). In all box plots, boxes represent the interquartile ranges (IQRs) between the first and third quartiles, and the line inside the box represents the median; whiskers represent the lowest or highest values within 1.5 × IQR from the first or third quartiles. Source data are provided as a Source Data file.

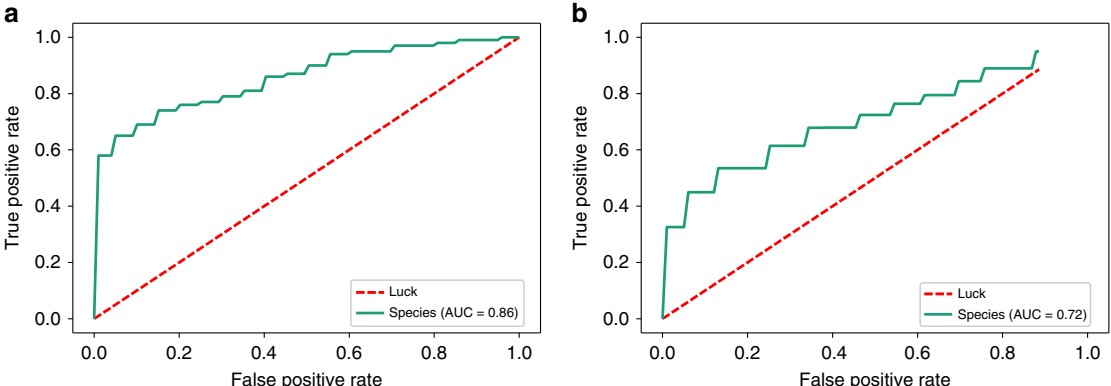

**Fig. 3 Gut microbial species differentiate UIA patients from controls. a** Receiver operating curve (ROC, turquoise) according to cross-validated random forest models on 47 species for the training set (the first cohort, controls, $n = 100$; UIA patients, $n = 100$). The area under the receiver operating curve (AUC) is 0.86, and the 95% confidence interval (CI) is 0.81–0.91. **b** ROC for the test set (the second cohort, controls, $n = 40$; UIA patients, $n = 40$). The AUC is 0.72, and the 95% CI is 0.65–0.79.

microbial pathways that were differentially abundant using both HUMAnN2 and KEGG. In particular, isoleucine biosynthesis and methionine biosynthesis were enriched in the controls. This result demonstrated a difference in the potential of the microbiota to produce branched-chain and aromatic amino acids between UIA patients and controls. This finding is consistent with the altered levels of these classes of amino acid in the microbiota of obese humans[16] and mice[17]. Overall, although functional annotation analyses are predictive, these data demonstrated that impairment of the UIA microbiome may evoke a disease-linked state through interference with physiological metabolic functions, especially fatty acid and amino acid metabolism.

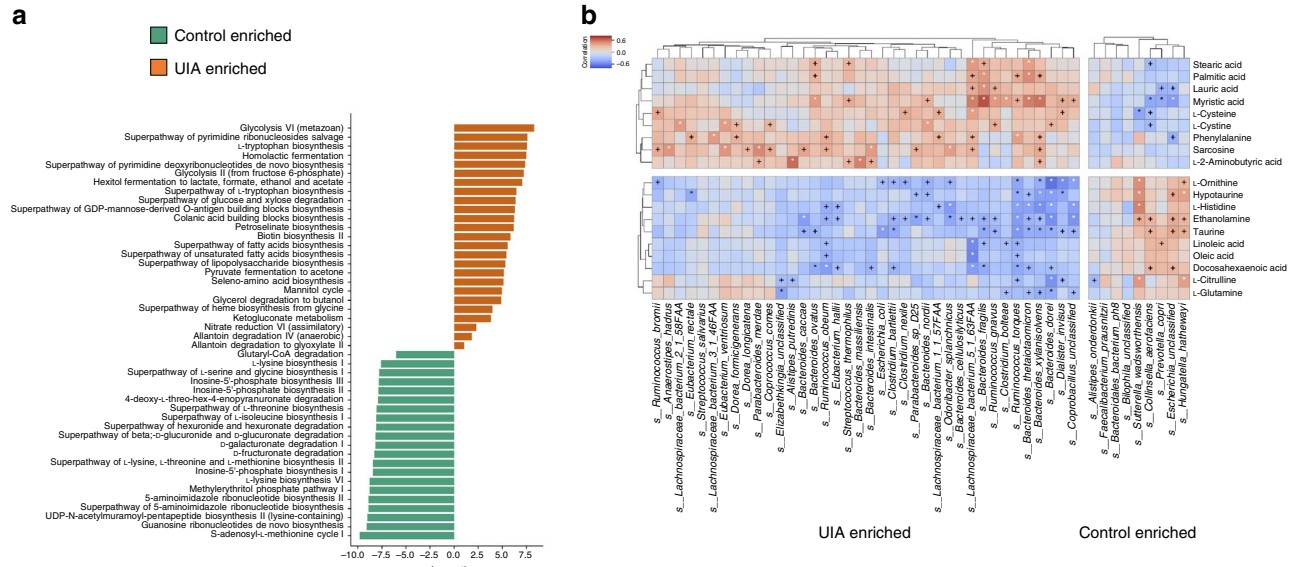

**Fig. 4 Functional characterization of the UIA microbiome. a** Pathways with biologically significant differential abundance between UIA patients and controls ($n = 100$). Turquoise and orange, control- and UIA-enriched pathways. **b** Heat maps of Spearman's rank correlation coefficients between 19 altered amino acids and fatty acids and 47 UIA-associated gut microbial species. $+P < 0.05$, $*P < 0.01$. Pairs with low correlations ($|r| < 0.4$) are not shown ($n = 30$). Source data are provided as a Source Data file.

**Associations of gut microbial species with circulating metabolites**. We functionally characterized the UIA microbiome based on the metabolism of fatty acids and amino acids by further exploring their metabolic profiles in the sera of a subset of 60 participants (30 UIA patients and 30 controls) from the first cohort using targeted metabolomics analysis. In total, the serum concentrations of 7 of 9 fatty acids and 12 of 32 amino acids differed substantially between UIA patients and controls (Supplementary Data 11). We then investigated whether the abundance of the 19 altered circulating metabolites correlated with the 47 altered gut microbiota at the species level. Spearman's correlation of differentially enriched species and metabolites was calculated, and a heat map was hierarchically clustered to represent the species-metabolite-associated patterns. Notably, consistent with the proposed differential capacities for producing aromatic amino acids in the microbiota of UIA patients, the serum concentrations of phenylalanine were considerably higher in UIA patients than in controls (Fig. 4b). Furthermore, we found that circulating amino acids such as taurine, hypotaurine, L-histidine and L-citrulline, which were significantly decreased in UIA patients, had a strong association with altered microbial species. Together, although it remains to be explored whether these metabolic products are directly metabolized by the gut microbiota, our data indicate that gut microbial species can modulate the levels of circulating amino acids and fatty acids that are further correlated with UIA.

**UIA gut microbiota contributes to the intracranial aneurysms**. To further determine whether changes in the gut microbiota are a direct factor in the progression of UIAs in vivo, fecal bacteria from UIA patients and controls were transplanted into mice. Depletion of the gut microbiota was achieved by administering a mouse broad-spectrum antibiotic cocktail. For fecal transplantation, mice were administered 100 μL of the stool supernatant from two individuals with UIAs and two control donors twice at a 1-day interval (Supplementary Table 1). The fecal samples of human donors and recipient mice post-transplantation were investigated by metagenomic shotgun sequencing. As expected, PCoA at the species level showed that the control donors were distributed near the center of gravity of 100 controls, and the UIA donors were distributed near the center of gravity of the 100 UIA patients (Supplementary Fig. 8a). Furthermore, to demonstrate the extent to which the mouse recipients had microbial profiles similar to the human donors, we used SourceTracker analyses to determine the extent of donor engraftment using a Bayesian algorithm[18]. We found that 61.5% of the fecal bacterial communities in mice treated with control feces were attributable to the control donors, and 53.8% of the fecal bacterial communities in mice treated with UIA feces were attributable to the UIA donors (Supplementary Fig. 8b; Supplementary Data 12 and 13).

To further identify the microbial species after fecal transplantation, MetaPhlAn2 was run to identify the taxonomic abundances at the species level in mice. Among the 14 differentially abundant species between mice treated with UIA patient feces and those treated with control feces, 5 species (35.7%), namely, *H. hathewayi*, *O. splanchnicus*, *A. putredinis*, *B. intestinalis*, and *B. nordii*, were also differentially enriched in MetaPhlAn2-associated species in UIA patients compared with controls in the two cohorts (Supplementary Data 14). The relative abundances of the top five most abundant species that showed significant differences between the two groups are shown in Supplementary Fig. 9.

In all, 20 mice treated with UIA patient feces and 20 treated with control feces underwent aneurysm-induction surgery. There was no mortality during the surgical procedure. Compared with mice treated with control feces (Con-FMT), treatment with feces from UIA patients (UIA-FMT) significantly increased the overall incidence of aneurysms (Fig. 5a, b and Supplementary Fig. 10a; UIA-FMT vs Con-FMT: 85% vs 45%; $n = 20$; $P = 0.019$) and the rupture rate (Fig. 5b and Supplementary Fig. 10b; UIA-FMT vs Con-FMT: 82% vs 22%; $n = 17$ vs 9; $P = 0.009$). Moreover, a symptom-free curve (Kaplan–Meier analysis) was plotted after excluding mice that did not have intracranial aneurysms (Fig. 5c). The log-rank test revealed a significant increase in aneurysmal rupture with UIA fecal treatment ($P = 0.0056$). Together, these findings provide direct evidence that the gut microbiota can influence the formation and rupture of intracranial aneurysms in the host.

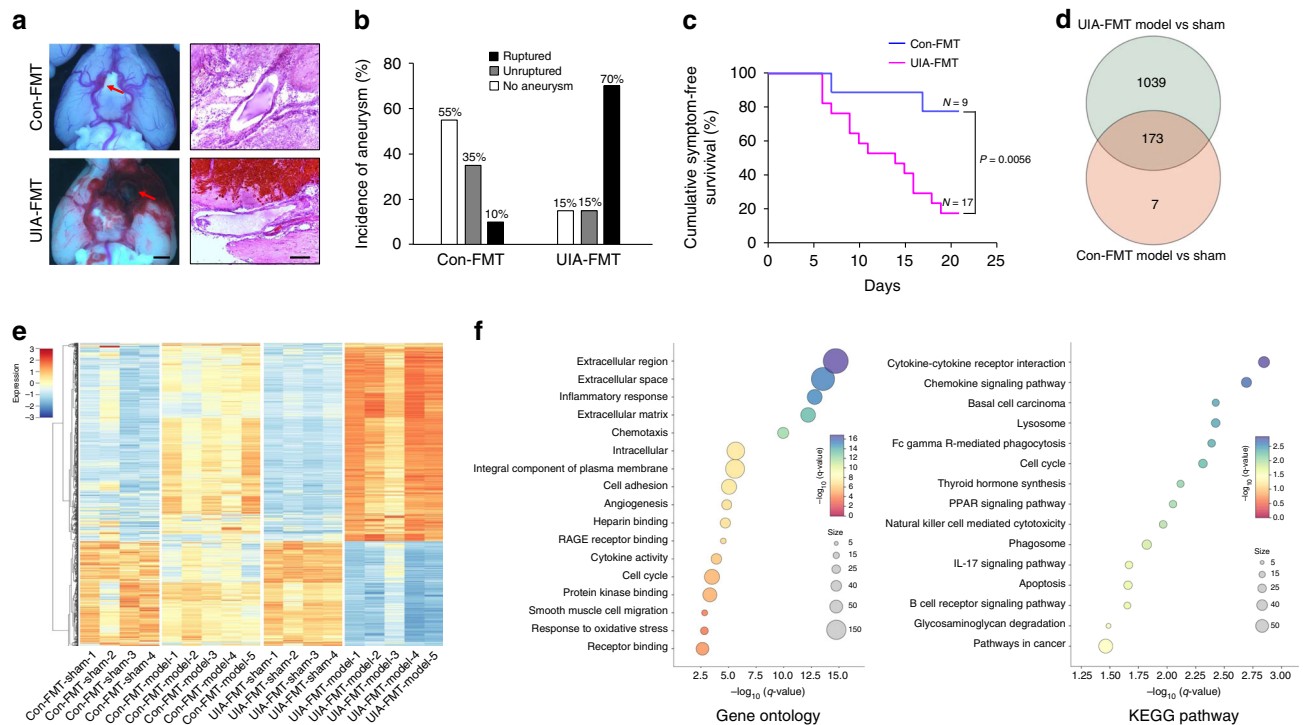

**Fig. 5 UIA gut microbiota contributes to the intracranial aneurysms. a** Representative images of intracranial aneurysms (left; arrows, aneurysms; scale bar, 1 mm) and cerebral arteries stained with hematoxylin-eosin (right; scale bar, 100 μm) in mice that received fecal microbiota transplantation (FMT) from UIA patients or controls, followed by aneurysm induction with angiotensin II and elastase. Independent experiments were repeated at least five times. **b** Incidence of unruptured and ruptured aneurysms 21 days after induction (n = 20). **c** Cumulative symptom-free curves for mice with aneurysms showing the time course of the onset of symptoms (n = 9 for control-FMT group, n = 17 for UIA-FMT group; log-rank (Mantel–Cox) test). **d** Venn diagram of the number of DEGs. Number in cyan circle, DEGs between the aneurysm induction group and the sham group after UIA-FMT. Number in pink circle, DEGs between the aneurysm induction group and the sham group after Control-FMT. **e** Right, heat maps of cluster analysis showing the transcript expression levels of DEGs in the cerebral vessels of the control-FMT and UIA-FMT groups on day 5 after aneurysm induction. The color scale illustrates the relative expression levels across all samples: orange, above the mean; blue, below the mean. Left, dendrogram showing the clustering of transcripts (n = 4 for sham groups, n = 5 for aneurysm induction groups). **f** Top terms showing enrichment based on −log$_{10}$ (q-value) from GO enrichment analysis (**left**) and KEGG pathway analysis (**right**) of the 1039 UIA-FMT-induced DEGs. Source data are provided as a Source Data file.

**UIA gut microbiota aggravates gene transcriptional alterations.** To gain mechanistic insights into the effects of the UIA gut microbiota on the formation and rupture of intracranial aneurysms, we performed whole-transcriptomic analysis of cerebral vessels using RNA-seq. In mice transplanted with feces from UIA patients, a total of 1212 genes (850 upregulated and 362 downregulated) were differentially expressed between the sham group and the angiotensin II- and elastase-treated group (Fig. 5d, e, Supplementary Data 15). In contrast, in mice transplanted with control feces, only 180 genes were differentially expressed between these two groups. Among these 180 genes, 173 overlapped between the transplantation of feces from UIA patients and controls. To more specifically elucidate the underlying mechanism by which the UIA gut microbiota affected intracranial aneurysms, we excluded these 173 genes from the 1212 differentially expressed genes (DEGs) induced by UIA gut microbiota, i.e., 1039 genes were used for further functional analysis. We investigated the intersection between these 1039 genes and another human transcriptome-wide characterization of gene expression associated with UIA (GSE26969) and found considerable overlap between DEGs across species (Supplementary Fig. 11, Supplementary Data 16). This suggested that the profound alterations of the transcriptional profile evoked by the UIA gut microbiota are a shared feature of intracranial aneurysms. Furthermore, this list of 1039 genes was subjected to Gene Ontology enrichment analysis to identify overrepresented biological functions and canonical pathways. These genes belonged to

multiple categories associated with inflammatory processes, cell adhesion, response to oxidative stress, and the extracellular matrix, which are known pivotal mechanisms of intracranial aneurysm formation and rupture (Fig. 5f)[19,20]. KEGG pathway analysis suggested that the DEGs were enriched in several important intracranial aneurysm-related signaling pathways, including apoptosis and several inflammation-related pathways, such as cytokine–cytokine receptor interactions and chemokine signaling (Fig. 5f). Taken together, the RNA-seq data revealed that the UIA gut microbiota significantly aggravated the profound changes in the gene transcriptional profile of cerebral vessels and that the mechanism by which intracranial aneurysms are induced by UIA gut microbiota may involve inflammation, modulation of the extracellular matrix in cerebral vessels, and apoptosis.

**Taurine reduces the incidence of intracranial aneurysms.** To validate the altered circulating metabolites in UIA patients and to further assess the relationships between gut microbiota and circulating metabolites, we performed targeted metabolomics profiling of fatty acids and amino acids in mice after fecal transplantation. In total, the serum concentrations of 2 of 8 fatty acids and 8 of 38 amino acids differed substantially between mice transplanted with feces from UIA patients and controls (Supplementary Data 17). Specifically, among these altered metabolites, taurine, L-histidine, and linoleic acid also corroborated the lower abundance in UIA patients compared with controls (Supplementary Fig. 12). On this basis, we next determined whether

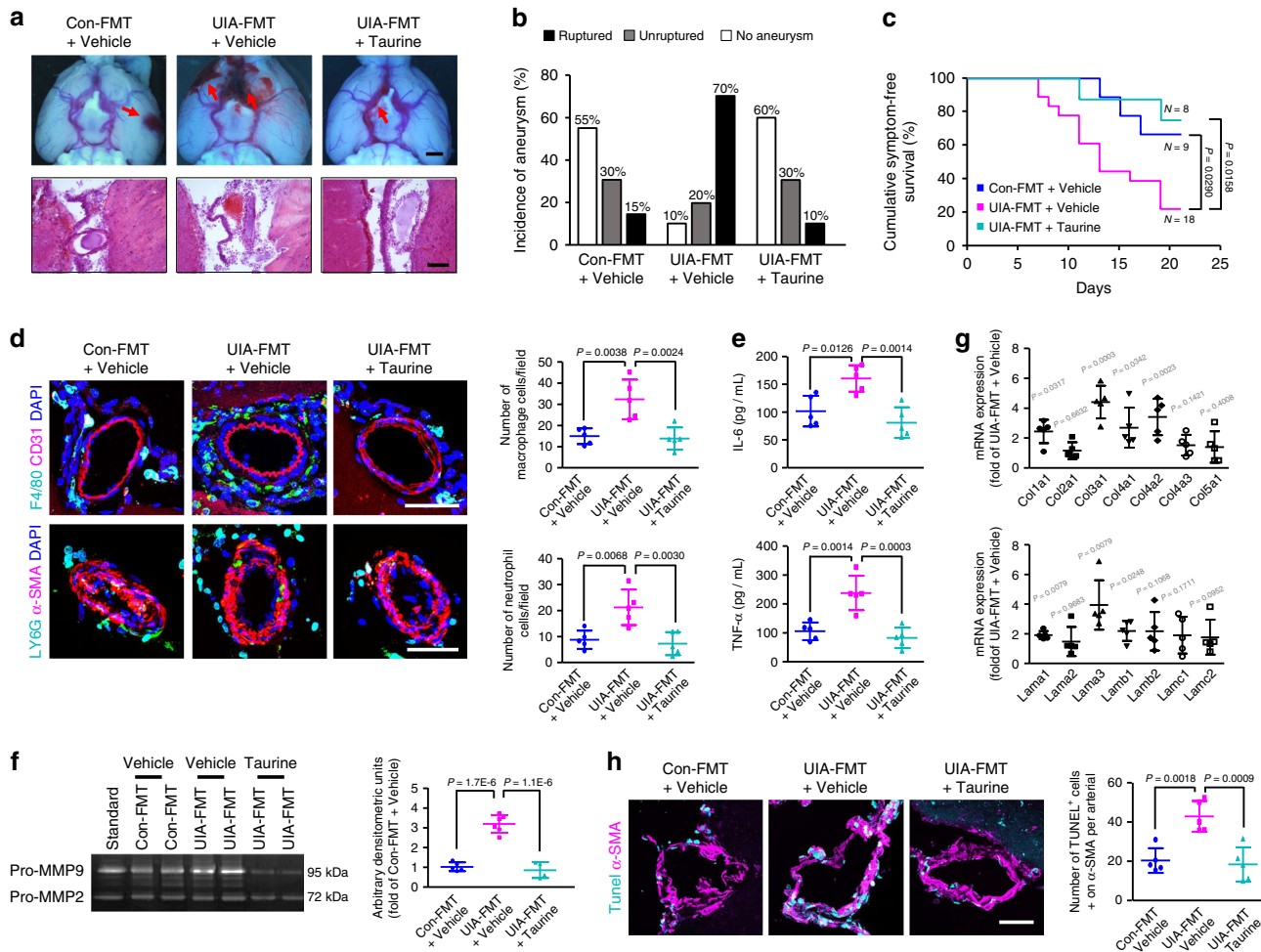

**Fig. 6 Taurine reduces the incidence of intracranial aneurysms. a** Representative images of intracranial aneurysms (upper; arrows, intracranial aneurysms; scale bar, 1 mm) and cerebral arteries stained with hematoxylin-eosin (lower; scale bar, 100 μm) in mice that received fecal microbiota transplantation (FMT) from UIA patients or controls, followed by taurine supplementation and aneurysm induction with angiotensin II and elastase. Independent experiments were repeated at least five times. **b** Incidence of unruptured and ruptured aneurysms 21 days after aneurysm induction ($n = 20$). **c** Cumulative symptom-free curves for mice with aneurysms to show the time course of symptom onset ($n = 9$ for control-FMT + Vehicle group, $n = 18$ for UIA-FMT + Vehicle group, $n = 8$ for UIA-FMT + Taurine group; log-rank (Mantel–Cox) test). **d** Upper panels, representative images and quantification of F4/80⁺ macrophage immunostaining surrounding CD31⁺ cerebral vessels. Lower panels; representative images and quantification of Ly6G⁺ neutrophil immunostaining surrounding α-SMA⁺ cerebral vessels (scale bars, 50 μm). **e** Levels of IL-6 and TNF-α in serum measured by ELISA. **f** Representative gelatin zymography and quantitative analysis of the MMP-9 activity in cerebral vessels from each group ($n = 6$; one-way ANOVA with the Bonferroni post hoc test). **g** Quantitative PCR showing the expression of pivotal components of arterial structure ($n = 5$; Student's unpaired two-tailed t-test or Mann–Whitney U test with the exact method). **h** Representative images and quantification of α-SMA⁺ vascular smooth muscle cell apoptosis in each group (scale bar, 50 μm). Data are the mean ± SD. For (**d**), (**e**), and (**h**), $n = 5$; one-way ANOVA with the Bonferroni post hoc test. Source data are provided as a Source Data file.

supplementation with taurine, L-histidine, or linoleic acid has potential therapeutic value for reducing the formation and rupture of intracranial aneurysms in mice.

Compared with UIA fecal treatment, taurine supplementation normalized the serum levels of taurine (Supplementary Fig. 13) and significantly reduced the overall incidence of aneurysms (Fig. 6a, b and Supplementary Fig. 14a; UIA-FMT + Taurine vs UIA-FMT + Vehicle: 40% vs 90%; $n = 20$; $P = 0.002$) and the rupture rate (Fig. 6b and Supplementary Fig. 14b; UIA-FMT + Taurine vs UIA-FMT + Vehicle: 25% vs 78%; $n = 8$ vs 18; $P = 0.026$). Kaplan–Meier analysis further revealed a significant decrease in aneurysmal rupture with taurine supplementation ($P = 0.0158$, Fig. 6c). Notably, taurine supplementation did not affect blood pressure, indicating that an antihypertension-independent mechanism is involved in the protective effects of taurine (Supplementary Fig. 15).

Intracranial aneurysms are increasingly recognized as a condition driven by chronic inflammation[21,22]. We observed cumulative infiltration of macrophages and neutrophils in perivascular spaces in mice transplanted with feces from UIA patients (Fig. 6d). Together with increased levels of the pro-inflammatory cytokines interleukin (IL)-6 and tumor necrosis factor alpha (TNF-α) (Fig. 6e), these data confirmed that stronger vascular inflammation was evoked by the UIA gut microbiota. Notably, inflammatory processes were dramatically ameliorated by supplementation with taurine. It has been reported that the inflammatory process in cerebral arteries leads to matrix metalloproteinase (MMP)-mediated degradation of the extracellular matrix and apoptosis of smooth muscle cells[19,23]. Using gelatin zymography, we confirmed that the MMP-9 activity in cerebral arteries was significantly induced by treatment with feces

from UIA patients compared with treatment with control feces (Fig. 6f). Likewise, the mRNA levels of collagen IV and laminin, which play substantial roles in arterial structure, were significantly decreased after UIA fecal treatment (Fig. 6g). Importantly, both MMP activation and extracellular matrix remodeling were dramatically ameliorated by taurine supplementation. Moreover, both significantly reduced TUNEL-positive vascular smooth muscle cell apoptosis and compensatory hypertrophy with wall thickening of cerebral arteries were found after taurine supplementation (Fig. 6h). However, we did not find a protective role of L-histidine or linoleic acid in reducing the incidence of the formation and rupture of intracranial aneurysms (Supplementary Fig. 16). Taken together, these data clearly demonstrated the essential roles of taurine in blunting inflammatory processes, reducing extracellular matrix remodeling, and maintaining the structural integrity of cerebral arteries, all of which ultimately reduced the formation and rupture of intracranial aneurysms.

To further corroborate the critical role of taurine reduction in intracranial aneurysm pathogenesis, mice that transplanted with feces from control donors were supplemented with 3% β-alanine in the drinking water, which significantly decreased the plasma concentration of taurine (Supplementary Fig. 17a). Compared with Vehicle treatment, β-alanine treatment significantly increased the overall incidence of aneurysms and the rupture

rate (Supplementary Fig. 17c–e). Kaplan–Meier analysis further revealed a significant increase in aneurysmal rupture with β-alanine administration (Supplementary Fig. 17f). These data demonstrated that the diminished level of taurine increases the incidence of the formation and rupture of intracranial aneurysms in mice.

**H. hathewayi protects against the intracranial aneurysms.** To further investigate the extent to which the altered microbiome in UIAs is associated with the changed circulating taurine level in the host, we calculated Spearman's rank correlation between altered metabolites and altered species in mice after fecal transplantation. We found that five species were positively correlated and 1 species was negatively correlated with taurine levels (Fig. 7a). Among these 6, H. hathewayi was the only overlapping species that also positively correlated with taurine levels in the human study. Furthermore, lower relative abundances of H. hathewayi all occurred in UIA patients in the two cohorts and mice treated with UIA patient feces compared with their respective controls (Fig. 7b). On this basis, to explore the possible causal role of H. hathewayi in the regulation of the level of circulating taurine and the formation and rupture of intracranial aneurysms, mice were gavaged with live H. hathewayi or vehicle [sterile phosphate-buffered saline (PBS)] after

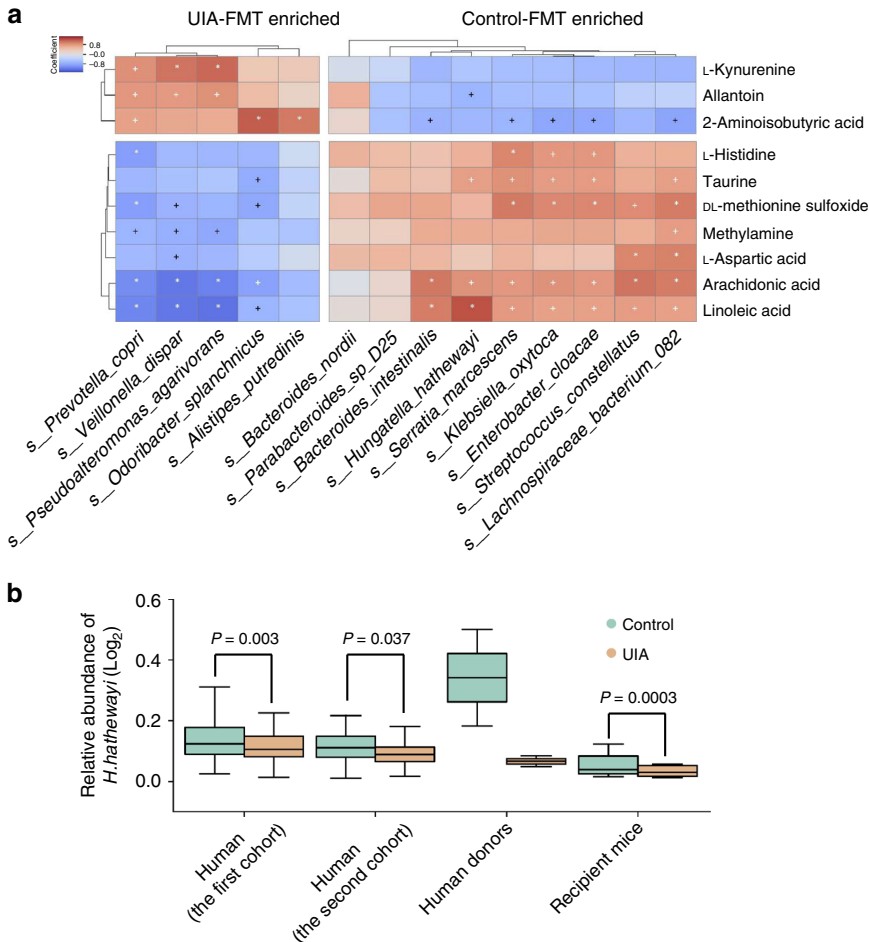

**Fig. 7 Associations of gut microbial H. Hathewayi with circulating taurine in mice. a** Heat maps of Spearman's rank correlation coefficients between 10 altered metabolites and 14 differentially abundant gut microbial species. $^{+}P < 0.05$, $^{*}P < 0.01$. Pairs with low correlations ($|r| < 0.4$) are not shown ($n = 10$). **b** Relative abundances of H. hathewayi in two cohorts of UIA patients ($n = 100$; $n = 40$), Human donors ($n = 2$), and mice that received fecal microbiota transplantation (FMT) from UIA patients or controls ($n = 10$). Two-tailed Wilcoxon rank-sum test. In all box plots, boxes represent the interquartile ranges (IQRs) between the first and third quartiles, and the line inside the box represents the median; whiskers represent the lowest or highest values within 1.5 × IQR from the first or third quartiles.

fecal transplantation. Real-time PCR analysis showed a significantly reduced amount of *H. hathewayi* in the feces of mice treated with UIA patient feces compared with that in the feces of mice treated with control feces. Oral gavage with $1 \times 10^9$ colony-forming units of live *H. hathewayi* was sufficient to restore the diminished level caused by the transplantation of UIA patient feces (Fig. 8a). Importantly, the plasma concentration of taurine also significantly increased after *H. hathewayi* gavage (Fig. 8b). Then, 3 groups of mice underwent aneurysm-induction surgery (representative intracranial aneurysms in the 3 groups are shown in Fig. 8c). Compared with UIA fecal treatment, *H. hathewayi* gavage significantly reduced the overall incidence of aneurysms (Fig. 8d

and Supplementary Fig. 18a; UIA-FMT + *hathewayi* vs UIA-FMT + Vehicle: 45% vs 90%; $n = 20$; $P = 0.006$) and the rupture rate (Fig. 8d and Supplementary Fig. 18b; UIA-FMT + *hathewayi* vs UIA-FMT + Vehicle: 44% vs 83%; $n = 9$ vs 18; $P = 0.026$). Kaplan–Meier analysis further revealed a significant decrease in aneurysmal rupture with *H. hathewayi* gavage ($P = 0.0134$, Fig. 8e). Furthermore, inflammatory processes, extracellular matrix remodeling, and MMP-9 activity in cerebral arteries were all dramatically ameliorated by *H. hathewayi* gavage (Fig. 8f–i). Likewise, *H. hathewayi* gavage significantly reduced TUNEL-positive vascular smooth muscle cell apoptosis in cerebral arteries (Fig. 8j). Taken together, these findings provide direct evidence

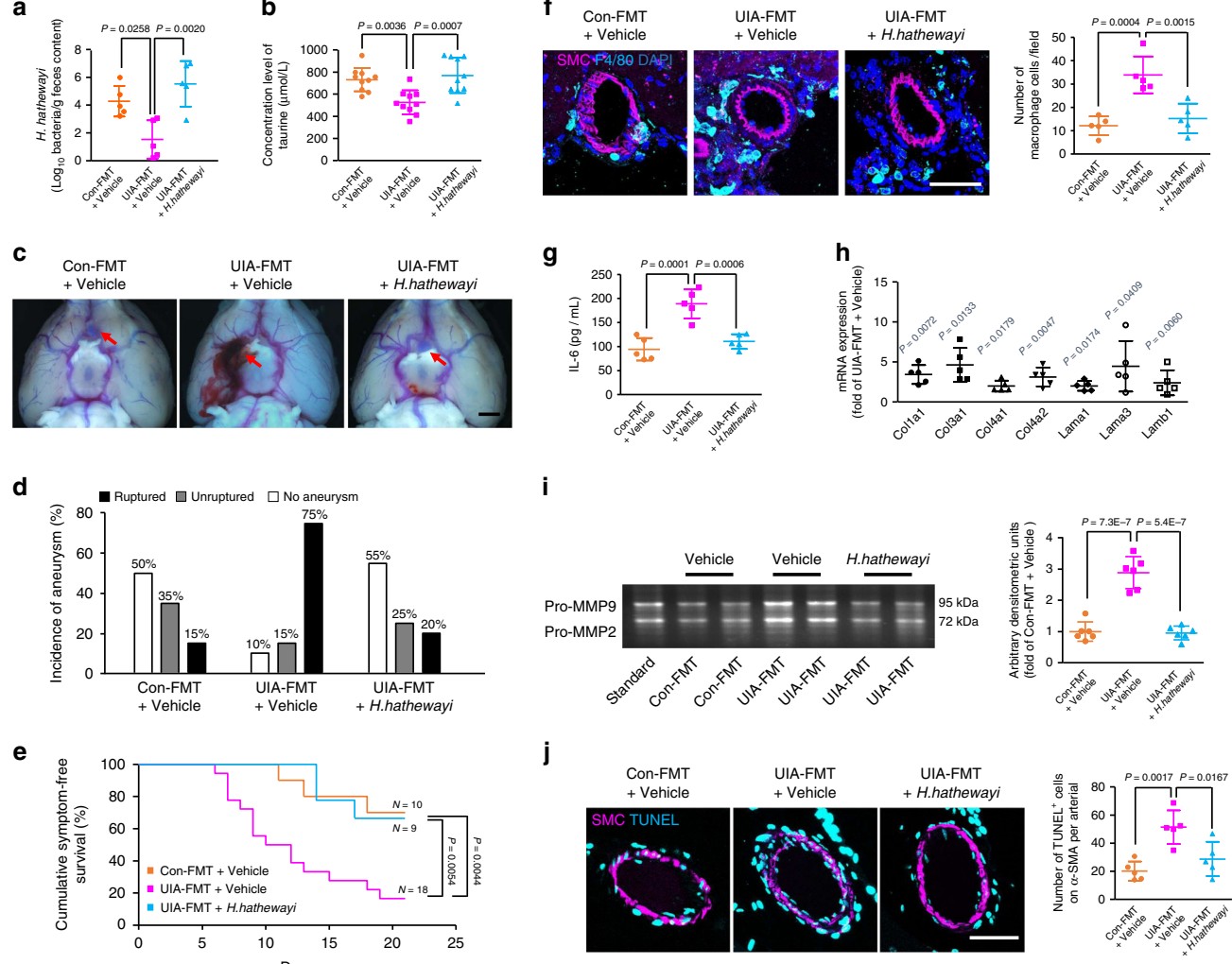

**Fig. 8 *H. hathewayi* protects against the intracranial aneurysms. a** Quantitative PCR detection of *H. hathewayi* in fecal DNA in each group ($n = 5$). **b** Serum concentrations of taurine in each group measured by UHPLC-MS/MS analyses ($n = 10$; one-way ANOVA with the Bonferroni post hoc test). **c** Representative images of intracranial aneurysms in UIA patients and controls that received fecal microbiota transplantation (FMT), followed by oral gavage with *H. hathewayi* and aneurysm induction with angiotensin II and elastase (arrows, intracranial aneurysms; scale bar, 1 mm). **d** Incidence of unruptured and ruptured aneurysms 21 days after aneurysm induction ($n = 20$). **e** Cumulative symptom-free curves for mice with aneurysms to show the time course of symptom onset ($n = 10$ for control-FMT + Vehicle group, $n = 18$ for UIA-FMT + Vehicle group, $n = 9$ for UIA-FMT + *H. hathewayi* group; log-rank (Mantel-Cox) test). **f** Representative images and quantification of F4/80[+] macrophage immunostaining surrounding CD31[+] cerebral vessels (scale bar, 50 μm). **g** Concentration of IL-6 in serum measured by ELISA. **h** Quantitative PCR showing the expression of pivotal components of arterial structure ($n = 5$; Student's unpaired two-tailed t-test or Mann–Whitney U test with the exact method). **i** Representative gelatin zymography and quantitative analysis of the MMP-9 activity in cerebral vessels from each group ($n = 6$; one-way ANOVA with the Bonferroni post hoc test). **j** Representative images and quantification of α-SMA[+] vascular smooth muscle cell apoptosis in each group (scale bar, 50 μm). Data are the mean ± SD. For (**a**), (**f**), (**g**), and (**j**), $n = 5$; one-way ANOVA with the Bonferroni post hoc test. Source data are provided as a Source Data file.

that *H. hathewayi* affects circulating taurine concentrations and protects mice against the formation and rupture of intracranial aneurysms.

## Discussion

Despite extensive research in recent years, relatively little is known about the precise mechanisms that contribute to the pathogenesis of UIAs. In addition to several untreatable risk factors, including genetic factors, old age, and female sex, there are also treatable risk factors that increase the occurrence of UIA, such as hypertension, cigarette smoking, and heavy alcohol use[24,25]. Accumulating evidence has shown that the gut microbiota is a pivotal risk factor in cardiovascular diseases by influencing host metabolism and immune homeostasis. However, no direct evidence has established a direct and causal relationship between an altered gut microbiota and UIAs. In the present study, we identified UIA-associated microbial species and explored their effects on host amino acid and fatty acid profiles. Using a mouse model of fecal transplantation and gavage with *H. hathewayi*, we uncovered evidence for the potential effects of altered *H. hathewayi* abundance on circulating taurine levels and on the increased occurrence of UIAs.

The current understanding characterizes chronic inflammation as the leading factor in the pathogenesis of UIAs[10,19]. In the present study, increased levels of the pro-inflammatory cytokines IL-6 and TNF-α were measured in the serum of mice transplanted with feces from UIA patients, corroborating a possible causal relationship between the altered gut microbiota of UIA patients and stronger systemic inflammation. Researchers have previously highlighted that *Bacteroides* are potentially pro-inflammatory and associated with inflammatory bowel disorder[26]. Interestingly, as shown by our data, a bacterial community of *Bacteroides* species was overrepresented in UIA individuals and mice treated with feces from UIA patients. Moreover, a recent study has reported that mice treated with *H. hathewayi* exhibit reduced release of cytokines and lower NF-κB activation in dendritic cells[27].

The gut microbiota is known to influence host intestinal homeostasis, in part via metabolic pathways such as short-chain fatty acid metabolism, bile acid metabolism, and amino acid metabolism. Concomitant with the significant change in gut microbial composition, we found changes in the gene functions of bacteria from UIA patients, especially in fatty acid and amino acid metabolism. We further performed metabolic profiling and found the altered circulating metabolites exhibited varying degrees of correlation with the gut microbial species that differed in abundance between UIA patients and controls and may exert pivotal influences on the pathophysiology of UIA development. Among the significantly altered metabolites, stearic acid was found to accumulate to high levels in the thickened aneurysmal wall and may be involved in the phenotypic switching of medial vascular smooth muscle cells of the aneurysmal wall[28]. In addition, taurine showed a considerable decrease in UIA patients compared with controls. Taurine has been shown to reduce inflammatory injury in various diseases and to play a protective role in acute ischemic stroke[29], traumatic brain injury[30], subarachnoid hemorrhage[31], and aortic aneurysm formation[32]. Depletion of the taurine transport system has been reported in the microbiome in atherosclerotic cardiovascular disease[7]. In the present study, taurine was also decreased in the serum of mice after transplantation of feces from UIA patients, implying that the altered gut microbiota of UIA patients may be a direct factor in the low level of taurine, not merely an adverse complication. Importantly, taurine supplementation significantly alleviated the formation and rupture of intracranial aneurysms in mice. The mechanistic findings involved blunted inflammatory processes, reduced extracellular matrix

remodeling, and maintenance of the structural integrity of cerebral arteries, all of which are pivotal mechanisms of intracranial aneurysm formation and rupture[19,20,23]. Interestingly, our results are analogous to those of a previous study that found that the circulating concentration of taurine is reduced in germ-free mice receiving fecal transplantation from individuals with autism spectrum disorder, and taurine supplementation improves repetitive and social behaviors in mice[33]. Nevertheless, the microbiome features of autism spectrum disorder depicted in this study are distinct from those of UIAs in the present study.

Possible UIA is increasingly identified in clinical practice as the use of MRI and CT becomes more common; magnetic resonance, digital subtraction, or CT angiography is then necessary to clarify the presence and specify the location of the aneurysm[34]. However, these techniques usually cannot achieve the high sensitivity required to detect aneurysms smaller than 4 mm[34]. In addition, these techniques are relatively expensive and are not available to patients for whom contrast administration is contraindicated. In the current study, we were able to identify UIA patients based on the differentially abundant species. These results demonstrate the presence of UIA-associated features in the gut microbiota that may be further developed into noninvasive and sensitive biomarkers for the early detection of UIAs, especially in the high-risk population without aneurysmal mass-effect symptoms.

Although endovascular treatment and surgical clipping serve as interventional options for UIAs, the optimal management strategy remains controversial because of the risk associated with these treatments[34]. A recent study reported that depletion of the gut microbiota with an antibiotic cocktail reduces the incidence of intracranial aneurysms in mice, representing a potential strategy for therapeutic intervention for UIAs[10]. Although the elimination of disease-causing microbiota by antibiotics is an obvious approach, the general consensus is that beneficial microbiota may have protective effects, and such a nonspecific antimicrobial approach may do more harm than good[35]. In the present study, we found that *H. hathewayi* was the only overlapped species that positively correlated with taurine levels in both human and mouse studies. Gavage with live *H. hathewayi* after fecal transplantation normalized the serum levels of taurine and protected mice against the formation and rupture of intracranial aneurysms, which indicated a causal role of *H. hathewayi* in the regulation of taurine and intracranial aneurysms. *H. hathewayi* is a strictly anaerobic bacterium that was described as a new species in 2001[14]. This bacterium is known to be responsible for the production of acetate, ethanol, carbon dioxide, and hydrogen, and there is no direct evidence that it is associated with taurine synthesis. Thus, we speculate that the effects of *H. hathewayi* on modulating the taurine levels may depend on interactions with certain additional microbial species. Further studies are clearly warranted to delineate how *H. hathewayi*, in concert with other microbial species, modulates taurine metabolism and whether *H. hathewayi* affects UIAs only via taurine. However, our data may add a dimension to our understanding of the beneficial effects of *H. hathewayi* on human health. Supplementation with *H. hathewayi*, or replacement with taurine, may serve as a viable and convenient approach for future investigations of UIA prevention and intervention.

Our study population consisted only of middle-aged to elderly Chinese individuals in a northern Chinese population. Thus, our current results potentially face limitations in generalizability. In future studies, the scope could be broadened to include other ethnicities within Asia and other groups such as Caucasians.

In summary, we describe the disordered profiles of the gut microbiota in UIA patients. Our findings extend our insights to a potential role of aberrant gut microbiota in the pathogenesis of UIAs and provide important evidence for a role of *H.*

*hathewayi*-associated taurine depletion as a key factor in the pathogenesis of UIAs. Our findings point toward a strategy for preventing the development of UIAs by supplementation with *H. hathewayi* and taurine.

## Methods

Data, analytical methods, and study materials for the purposes of reproducing the results or replicating the procedures will be made available upon request to the corresponding author, who manages the information.

**Study cohorts and patient characteristics**. The case-control study protocol was reviewed and approved by the Human Ethics Committee, Fuwai Hospital (Approval No. 2016-732), and the study was conducted in accordance with the Principles of Good Clinical Practice and the Declaration of Helsinki. Written informed consent was provided by all study participants or their legal proxies.

A total of 140 UIA patients and 140 age-, sex- and blood pressure-matched controls were collected from 2 cohorts. In the first cohort, individuals enrolled between April 2016 and November 2016 formed the discovery phase. A total of 100 UIA patients were consecutively enrolled from among those admitted to Beijing Tiantan Hospital. Thirty-seven controls were consecutively enrolled from Cangzhou Central Hospital, and another 63 controls were enrolled from a cohort who received biennial medical examinations at Kailuan General Hospital. Note that the metagenomic sequencing data from the 63 fecal samples from the latter controls were available from our previous study and were used in the present study (Supplementary Data 1)[5]. In the second cohort, individuals enrolled between July 2016 and December 2017 constituted the validation phase. Forty UIA patients were consecutively enrolled from among those admitted to Chinese PLA General Hospital and Beijing Tiantan Hospital. Forty controls were enrolled from Cangzhou Central Hospital and Tsinghua University Hospital.

Individuals who met the following criteria were considered eligible for the study: patients diagnosed with UIAs confirmed by either digital subtraction angiography or magnetic resonance angiography and who underwent microsurgical clipping. The exclusion criteria were the following: a family history of intracranial aneurysm, ongoing infectious diseases, inflammatory bowel diseases, irritable bowel syndrome, autoimmune diseases, liver diseases, renal diseases, cancer, and diarrhea, and the use of antibiotics or probiotics within 2 months of sample collection. The samples from UIA patients were collected prior to the microsurgical clipping. Patients who were treated for their UIAs before sample collection were also excluded. Moreover, controls were included from outpatients with minor illnesses and matched to UIA patients with respect to age, sex and blood pressure. Controls were free of aneurysmal symptoms or a history of subarachnoid hemorrhage following the same exclusion criteria as used for UIA patients. Demographic and clinical characteristics were obtained by face-to-face surveys and from hospital and medical examination records (Supplementary Tables 2 and 3).

**Biochemical measurements**. Human blood samples for clinical chemistry analyses were collected after an overnight fast for at least 10 h. Serum samples were centrifuged and stored at −80 °C until analysis. Fasting glucose and lipid profiles, including total cholesterol, high-density lipoprotein cholesterol, low-density lipoprotein (LDL) cholesterol, and triglycerides, were measured using an autoanalyzer (Hitachi Labspect008).

**DNA extraction from fecal samples**. Fecal samples freshly collected from each participant were immediately frozen at −20 °C and transported to the laboratory with an ice pack. Bacterial DNA was extracted at Novogene Bioinformatics Technology Co., Ltd., using Tiangen kits according to the manufacturer's recommendations.

**Metagenomic sequencing and gene catalog construction**. All samples were paired-end sequenced on the Illumina HiSeq X Ten platform (insert size 350 bp, read length 150 bp) at Novogene Bioinformatics Technology Co., Ltd. Following quality control, the reads aligned to the human genome with SOAP2[36] (v2.21, parameters: -s 135, -l 30, -v 7, -m 200, -x 400) were removed. The remaining high-quality reads were then assembled using SOAP denovo[1] (v2.04, parameters: -d 1 -M 3 -R -u -F) with several k-mer values (from 49 to 87), and only the scaffolds with the longest N50 value remained. Clean reads were mapped against the scaffolds using SOAP2[36] (v 2.21, parameters: -m 200 -x 400 -s 119). The unmapped reads from each sample were merged and assembled using the same parameters. The scaftigs (i.e., continuous sequences within scaffolds) longer than 500 bp were used to predict genes with MetaGeneMark[37] (prokaryotic GeneMark-hmm v2.10). A non-redundant gene catalog was constructed with CD-HIT[38] (v4.5.8, parameters: -G 0 -aS 0.9 -g 1 -d 0 -c 0.95) with a sequence identity cutoff of 0.95 and a minimum coverage cutoff of 0.9 for the shorter sequences. The abundance of genes in each sample was calculated by counting the number of aligned reads (SOAP2, parameters: -m 200 -x 400 -s 119) and normalizing by gene length[39].

**Taxonomic annotation and abundance profiling**. Taxonomic assignments for the genes were obtained by aligning the integrated NR database using DIAMOND[40] (v0.7.9.58, default parameters except that -k 50 -sensitive -e 0.00001). The taxonomic level of each gene was determined by the lowest common ancestor-based algorithm[41] from the significant matches, which were defined by e-values ≤1e−5 as the best hit5[39]. The abundance of a taxonomic level was calculated by summing the abundances of genes annotated to that group. Furthermore, with the quality-controlled reads, bacterial taxa was assigned and quantified at the species level using the Metaphlan2 method for each sample[11,12]. The extent of transfer of human-associated bacterial taxa was determined using the default parameters of SourceTracker software version 2.0.1[18].

**Functional annotation and abundance profiling**. Functional assignments of the genes were obtained using DIAMOND (v0.7.9.58, default parameters except that -k 50 –sensitive -e 0.00001) with the KEGG (v20180101)[42], eggNOG (v4.5)[43], and CAZy databases (v20150704). Each protein was assigned to the KEGG orthology and CAZy families by the highest-scoring annotated hits containing at least one HSP scoring >60 bits[44]. The abundance of a KEGG orthology/module was calculated by summing the abundances of genes annotated to the same feature. Furthermore, functional profiling of abundance in microbial pathways was performed using HUMAnN2[15] with the Chocophlan and Uniref50 database[45] to implement KEGG orthologies.

**Rarefaction curve, α-diversity, and β-diversity**. Rarefaction analysis was performed with R (v2.15.3, vegan package) to assess the gene richness in UIA patients and controls. The Shannon index (α-diversity) was calculated with QIIME (v1.7.0) based on the genus and species profiles. β-diversity (between-sample diversity) was estimated by the Bray-Curtis dissimilarity index matrices using the ANOSIM method in the vegan R package.

**Coabundance gene groups**. Marker genes associated with UIA patients and controls were identified by comparing the abundance of each gene with a $q$-value <0.05 (Wilcoxon rank-sum test)[16]. Then, marker genes were clustered into groups based on their abundance variation across groups[46]. Clusters with >50 genes were defined as metagenomic linkage groups (MLGs). MLG abundance was calculated by the average gene depth signal and weighted by gene length. Taxonomic assignment of the MLGs was performed according to the taxonomy of tracer genes[47]. Briefly, assignment to species requires 90% of the genes in an MLG to align with the species' genome with 95% identity and 70% overlap of query. Assigning MLG to a genus requires 80% of its genes to align to the genome with 85% identity in both DNA and protein sequences. Others were considered 'Unclassified'.

**Co-occurrence network of MLGs**. The co-occurrence network based on Sparse Correlations of species with differential abundance was constructed using SparCC[48]. Briefly, SparCC is particularly designed to deal with compositional data, and Pseudo P-values were calculated as the proportion of simulated permuted data sets with a correlation at least as extreme as that computed for the original dataset[49]. Then, the corrected P-values were calculated by the Benjamin-Hochberg method, and the edges in the network were filtered with P < 0.01.

**Correlation analysis of gut microbial species and metabolites**. Spearman's correlation of differentially enriched species and metabolites was calculated using the scipy-stats package. Heat maps were hierarchically clustered to represent the species-metabolite-associated patterns based on the correlation distance. All analyses and visualizations were implemented in python (v2.7.9) with the numpy (v1.9.2), scipy (v0.15.1), and matplotlib (v1.4.3) packages.

**Prediction model**. A partial least squares-discriminant analysis (PLS-DA) classifier was trained for the prediction of altered species between UIAs and controls in the first cohort of 200 samples (training set) and external validation in the second cohort of 80 samples using the scikit-learn (v0.19.1) package in python (v2.7.9). Four-fold cross-validation within the training set was used to evaluate the performance of the predictive model and obtain more accurate curves. The cross-validation accuracy curves from 30 random trials of the 4-fold cross-validation were averaged. The predictive model was constructed using the most important variables, which were further applied for receiver operating curve analysis. The performance of the models was measured as accuracy (AUC) when applied to the validation set and as confidence intervals with the bootstrap procedures. All of the above procedures were performed in python (v2.7.9) with the scikit-learn package (v0.19.1).

**Feature selection**. The multivariate statistical analysis PLS-DA was performed to discriminate UIA samples from controls[50,51]. The number of MLGs responsible for the difference in the microbiome profile scan of UIA patients and controls was obtained based on the Monte Carlo cross-validation framework. The stability or

reliability of the partial regression coefficient can be defined as follows:

$$c_j = \frac{\beta_j}{s(\beta_j)}, j = 1, 2, \ldots, p,$$

$$s(\beta_j) = \left( \sum_{i=1}^{N} \frac{(\beta_{ij}-\beta_j)^2}{N-1} \right)^{1/2},$$

where $c_j$ is utilized in the conjunction of the addition of random variables and the original data, and $\beta_{ij}$ symbolizes the regression coefficient of the $j$th variables in the PLS model, which is built by the $i$th $M$ randomly chosen samples ($M$ is half of the samples). In practice, using the Monte Carlo method substantially reduces the computational complexity. Theoretically, fewer samples are selected randomly from the calibration samples, and more $N$ is needed. However, it has been proven that $N = n^2$ is general enough to improve the performance of the Monte Carlo strategy. The selected features were determined by the accuracy of the cross-validation based on the features having a $cj$ value larger than different cutoffs. Then, variable importance was used to evaluate the contribution of the selected MLGs to UIA compared with controls.

**Targeted metabolomics profiling of serum samples**. A total of 30 UIA patients and 30 controls from the first cohort were subjected to targeted metabolomics analysis. In each group, we randomly selected 11 from among 50 males, and 19 from among 50 females. In addition, 20 recipient mice transplanted with fecal contents from controls or UIA patients were also subjected to targeted metabolomics analysis. Whole blood samples were collected, and the serum was separated by centrifugation.

**Targeted amino acid profiling**. Amino metabolites were detected by HPLC-QqQ-MS/MS[52]. One hundred μL of the serum sample was mixed with 300 μL of cold methanol for protein precipitation. After 10 min of centrifugation (11060$g$, 4 °C), 10 μL of supernatant was mixed with NEM solution for trapping thiols through click reaction forming RSH-NEM adducts. Then we removed multiple NEM by adding tBBT, and reduced disulfide bonds by TCEP. Derivatization of amino metabolites by 5-AIQC, we could quantitative detect 5-AIQC-RSH-NEM adducts and 5-AIQC-RSH adducts in a single run. The mixed standards were derivatized by the same way.

The UPLC-QqQ-MS/MS system was made up of an Agilent 1290 UPLC coupled to an Agilent 6460 triple quadrupole mass spectrometer equipped with an Agilent Jet Stream electrospray ionization (ESI) source (Agilent Technologies, Inc. Santa Clara, CA, USA). Chromatography was carried out with a C18 column (Agilent Zorbax Eclipse XDB-C18 Rapid Resolution HD, 2.1 × 100 mm, 1.8 μm). Solvent A is 100% ultrapure water/0.1% formic acid, and solvent B is methanol /0.1% formic acid. The solvent program is: 1% B (0–2 min), 1–3.8% B (2–4 min), 3.8–22% B (4–8 min), 22–25% B (8–12 min), 25–60% B (12–13 min), 60–80% B (13–13.51 min), and 80–95% B (13.51–16 min). The flow rate is 0.6 mL/min and column temperature is 50 °C. The mass spectra of the 5-AIQC-tagged amino metabolites with MRM mode showed the derivatized group including ions at m/z 171 to identify. Data acquisition and analysis were performed with Mass Hunter software (B.08.00, Agilent Technologies, Inc. Santa Clara, CA). The concentrations of amino metabolites were calculated from the calibration curves were prepared by diluting mixed standards.

**Targeted free fatty acid profiling**. Free fatty acids were detected by HPLC-QqQ-MS/MS[53,54]. Ten μL of the serum sample was mixed with 10 μL of methanol containing 2 ng of heptadecanoic acid as internal standard. Added 5 μL of BHT as a stabilizer (antioxidant), 100 μL of ultrapure water, 250 μL of methanol and 12.5 μL of HCl solution for protein precipitation. We mixed well and centrifuged (3000$g$, 4 °C, 60 s) the solution after adding 750 μL of isooctane, then extracted free fatty acids from the upper layer in the biphasic solution, and repeated twice. The combined isooctane phases were evaporated to dryness under a N2 stream and derivatized with AMPP.

The UPLC-QqQ-MS/MS system was made up of an Agilent 1290 UPLC coupled to an Agilent 6460 triple quadrupole mass spectrometer equipped with an Agilent Jet Stream electrospray ionization (ESI) source (Agilent Technologies, Inc. Santa Clara, CA, USA). Chromatography was carried out with a C18 column (Agilent Zorbax Eclipse XDB-C18 Rapid Resolution HD, 2.1 × 100 mm, 1.8 μm). Solvent A is 100% ultrapure water /0.1% formic acid, and solvent B is acetonitrile /0.1% formic acid. The solvent program is: 30% B-95%B (0–12 min), 95% B (12–13 min). The flow rate is 0.5 mL/min and column temperature is 40 °C. The mass spectra of the AMPP-tagged free fatty acids with MRM mode showed the derivatized group including ions at m/z 169.0 and 183.0 to identify. Data acquisition and analysis were performed with Mass Hunter software (B.08.00, Agilent Technologies, Inc. Santa Clara, CA). The concentrations of free fatty acids were calculated from the internal standard C17:0 because odd-chain fatty acids must not occur in the Mammalian samples.

**Mouse study**. Animal experiments were approved by the Committee on the Ethics of Animal Experiments of Fuwai Hospital and complied with the National Institutes of Health's Guide for the Care and Use of Laboratory Animals; the manuscript adheres to the Animal Research: Reporting of In Vivo Experiments

(ARRIVE) guidelines. Male wild-type C57BL/6N mice, 10 weeks old, were purchased from the National Resource Center of Model Mice. Female mice were not used to avoid any influence of sex steroids. All mice were housed in a specific pathogen-free environment (temperature, 22 °C ± 2 °C; humidity, 55% ± 5%; 12-h light/12-hour dark cycle) and fed a rodent diet ad libitum. The mice were numbered, and the cages were placed in a random order on the shelves. Investigators who were blinded to the treatment groups evaluated the outcomes in all mice and performed the analysis. The number of mice used is labeled in each figure.

**Microbiota depletion**. Depletion of gut microbiota before fecal transplantation was achieved by administering mice a broad-spectrum antibiotic cocktail (1 g/L ampicillin, 1 g/L metronidazole, 1 g/L neomycin, and 0.5 g/L vancomycin) via autoclaved (tap) drinking water for 2 weeks[55]. The drinking water was replaced twice per week.

**Fecal transplantation**. For microbiota transplantation, fresh fecal samples were collected from 2 UIA patients and 2 controls (The age of these four donors ranged from 53 to 59 years old). Stool (200 mg) was dissolved in 10 ml saline, shaken for 3 min, and centrifuged for 3 min at 4 °C, and 2 ml of supernatant was collected. After 2 weeks of antibiotic treatment, the mice were given 100 μL of the above supernatant by gavage twice at a 1-day interval[56].

**Culture and administration of *H. hathewayi***. *H. hathewayi* (DSM-13479, Leibniz-Institut DSMZ GmbH, Braunschweig, Germany) was cultivated under anaerobic conditions in PYG medium (104b, Leibniz-Institut DSMZ GmbH, Braunschweig, Germany). The concentration of *H. hathewayi* was calculated by measuring the absorbance at a wavelength of 600 nm. One week after fecal transplantation, mice were gavaged with 100 μL of either live *H. hathewayi* ($1 \times 10^9$ colony-forming units/mouse in sterile PBS) or sterile PBS, three times a week. Twenty-one days after *H. hathewayi* gavage, fecal samples were collected and the bacterial DNA was extracted. Quantitative real-time PCR was performed to detect the abundance of *H. hathewayi*[57]. The results are presented as the log of per g fecal DNA. Then, the mice underwent aneurysm-induction surgery, and *H. hathewayi* gavage was continued three times a week until they developed neurological symptoms or until day 21 after surgery if the mouse did not have neurological signs.

**Mouse model of intracranial aneurysm**. One week after fecal transplantation, the right renal artery and the left common carotid artery of mice were ligated using an 8–0 nylon suture under anesthesia with 3% isoflurane in an O2 mixture (1 L/min). Sham groups of mice received only sham operation. One week later, the mice were anesthetized with 3% isoflurane in an O2 mixture (1 L/min) and fixed in a stereotaxic frame with a mouse adaptor (World Precision Instruments) to hold the skull horizontally. A Hamilton syringe with a 26 G needle (Hamilton, Reno, NV, USA) attached to a syringe pump (World Precision Instruments) was advanced to the right basal cistern using the coordinates from the Mouse Brain Atlas (2.7 mm rostral, 1.0 mm lateral to bregma, and 6.2 mm ventral to the skull surface). Elastase solution (35 mU, E7885, Sigma-Aldrich) was then injected at 0.2 μL/min. After elastase injection and closure of the skin over the burr hole, a separate small incision was made in the dorsal skin between the scapulae. A micro-osmotic pump (Durect Corp., CA, USA) containing angiotensin II (AngII; 1000 ng/kg/min; Sigma-Aldrich, MO, USA) in PBS was implanted into a subcutaneous pocket. Sham groups only received saline injections into the right basal cistern and implantation of a saline-filled osmotic pump[58,59].

To investigate whether taurine, L-histidine, or linoleic acid has significant implications for UIA progression and therapeutic intervention, one week after fecal transplantation, mice were fed taurine at 150 mg/day[60], L-histidine at 500 mg/day[61] or linoleic acid at 20 mg/day[62] by gavage until they developed neurological symptoms or until day 21 after pump implantation if the mouse did not have neurological signs. Vehicle groups of mice received only saline by gavage at the same time points.

To further corroborate the critical role of taurine production in intracranial aneurysm pathogenesis, mice were transplanted with feces from control donors after depletion of the gut microbiota by antibiotic cocktail. Then mice were supplemented with 3% β-alanine in the drinking water for 2 weeks[63], which has been reported to significantly decrease the plasma concentration of taurine in rodent. Then the mice underwent aneurysm-induction surgery and continued to drink β-alanine or Vehicle until they developed neurological symptoms or until day 21 after surgery if the mouse did not have neurologic signs.

Systolic blood pressure was measured in conscious mice on day 5 after pump implantation using tail-cuff plethysmography (BP-2010A, Softron, Japan). The mice were closely monitored each day for any neurological symptoms until the end of the study. If any neurological symptoms, such as inactivity, circling, paresis, or ≥15% weight loss were observed, the mice were immediately killed and perfused with PBS, followed by a gelatin-containing blue dye to visualize cerebral vessels. Aneurysms were defined as a localized outward bulging of the vascular wall with a diameter greater than that of the parent artery[64].

**Tissue collection**. Cerebral arteries (anterior and posterior cerebral arteries, anterior and posterior communicating arteries, middle cerebral arteries, and basilar

arteries) were isolated for RNA sequencing, quantitative real-time PCR, and gelatin zymography. Then, the brains were immersed in 4% paraformaldehyde for 6 h at 4 °C and cryoprotected in 20% sucrose at 4 °C overnight. Frozen coronal sections were cut at 20 μm. For hematoxylin-eosin staining, sections were incubated in hematoxylin for 2 min and eosin for 1 min.

**Immunohistochemistry**. Cerebral tissue sections were blocked with 1% bovine serum albumin and 5% normal donkey serum in PBS and were incubated with a combination of primary antibodies over night at 4 °C. The sections were then rinsed and incubated with Alexa Fluor 594-conjugated or Alexa Fluor 488-conjugated secondary antibodies, respectively. Nuclei were stained with DAPI. The following primary antibodies were used: goat anti-CD31 (1:400, AF3628, R&D Systems Inc., MN, USA), α-smooth muscle-Cy3™ antibody (1:1000, αSMA, clone 1A4, C6198, Sigma-Aldrich, USA), rat anti-Ly6G (1:200, 551459, BD Pharmingen, USA), and anti-F4/80 (1:200, ab6640, Abcam, MA, USA). The following secondary antibodies were used: Alexa Fluor 594-conjugated donkey anti-goat IgG (1:1000, ab150132, Abcam, MA, USA); Alexa Fluor 594-conjugated donkey anti-rat IgG (1:1000, ab150156, Abcam, MA, USA); Alexa Fluor 488-conjugated donkey anti-rat IgG (1:1000, ab150153, Abcam, MA, USA); Alexa Fluor 488-conjugated donkey anti-rabbit IgG (1:1000, ab150073, Abcam, MA, USA); Alexa Fluor 488-conjugated donkey anti-goat IgG (1:1000, ab150129, Abcam, MA, USA). All of the images were visualized under a Leica SP8 laser-scanning confocal microscope and a Leica DM6000B microscope. For the quantification of all markers, we chose the field with the highest number of cells around the middle cerebral artery for each sample and quantified the number of immunostained cells per high-power field (×40). Five mice per group were examined.

**TUNEL staining**. TUNEL staining (terminal deoxynucleotidyl transferase-mediated dUTP nick-end labeling, Sigma-Aldrich, MO, USA) was evaluated using an in situ cell death detection kit. Briefly, sections were incubated with α-SMA and secondary antibodies, followed by incubation with the TUNEL mixture according to the manufacturer's instructions. We assessed the numbers of TUNEL+ cells in α-SMA+ arteries using ImageJ software[65].

**Total RNA isolation**. Total RNA was extracted from the cerebral artery samples using the RNeasy Mini Kit with an on-column DNase step (Qiagen, Hilden, Germany) according to the manufacturer's protocol. Immediately following extraction, the total RNA concentration and A260:A280 ratio of each sample was determined on a NanoDrop 2000 (Thermo Fisher Scientific, IL, USA).

**Mouse RNA sequencing and functional analysis**. Total RNA was purified using an RNeasy micro kit (Qiagen, Hilden, Germany). The TruSeq RNA Sample Preparation Kit V2 (Illumina, San Diego, CA) was used for next-generation sequencing library construction according to the manufacturer's protocols. The cDNA product was amplified, and sequencing adapters and barcodes were ligated onto the fragments for each sample to create cDNA libraries ready for sequencing. Sequencing was performed by Annoroad Gene Technology Corp. (Beijing, China) using the Illumina HiSeq X Ten platform.

The clean data without ribosome RNA reads for each sample were aligned to the reference genome (ftp://hgdownload.cse.ucsc.edu:/goldenPath/mm10/chromosomes) using HISAT2 (v2.0.1)[66]. Then, four procedures, 'stringtie, stringtie—merge, cuffquant, and cuffnorm with default parameters, were used to reconstruct transcripts, identify novel transcripts, quantify transcripts, and normalize expression values (FPKM, Fragments per Kilobase of transcript per Million mapped reads)[67]. For differential expression analysis, the R language (v3.2.1) with the edgeR package was used to identify the DEGs[68]. The fold change between the two groups was calculated as logFC = log2 (experimental/control group). Genes in the two groups with |logFC| > 1 and q-value < 0.05 were defined as DEGs. DEGs were hierarchically clustered to represent the expression patterns using the ward method for the Euclidean distance matrix. The clustered gene expression profile is shown as the mean of log2 (FPKM) in each group. Interproscan (v5.8-49.0) and blast2go software were used to annotate the domains, gene families, and gene ontology (GO) functions. KOBAS (v2.0) was used to map the genes to KEGG pathways with default parameters[69]. A hypergeometric distribution test was carried out to identify GO functions and KEGG pathways in which DEGs were significantly enriched (q-value < 0.05) compared with total background expressed genes. Enriched GO items and KEGG pathways were plotted by python (v2.7.9) with the matplotlib (v1.4.3) package.

**Quantitative real-time PCR**. Relative quantification by real-time PCR was performed using SYBR Green to detect PCR products in real time with the ABI PRISM 7500 Sequence Detection System (Applied Biosystems). Melting curve analysis was performed at the end of each PCR. The sequences of oligonucleotide primers were designed according to the cDNA sequences in the GenBank database using Primer Express software (Applied Biosystems) and are listed in Supplementary Table 4.

**ELISA**. The plasma levels of TNF-α and IL-6 in mice were measured using ELISA kits (R&D Systems Inc., MN, USA). All values were in the linear range and were calculated based on known protein concentrations.

**Gelatin zymography**. Equal amounts of protein from the cerebral vessel samples were loaded and separated on a 10% Tris-glycine gel with 0.1% gelatin as the substrate. Then, the gel was washed and renatured with 2.5% Triton X-100 buffer. After incubation with developing buffer at 37 °C for 24 h, the gel was stained with 0.05% Coomassie R-250 dye (Sigma-Aldrich, MO, USA) for 30 min and de-stained. The gelatinase standard was mixed with pro-MMP-9 and pro-MMP-2 (Sino Biological Inc., Beijing, China)[70]. Pro-MMP-9 activity was evaluated by optical density.

**Statistics**. The Shannon index was calculated at the genus level with QIIME (Quantitative Insights Into Microbial Ecology). Bray-Curtis dissimilarity matrices were calculated and used for ordination by PCoA. These matrices were also used to assess differences in β-diversity by ANOSIM. PCoA was carried out for all dimension reduction analyses using the vegan package in R software. P-values were corrected for multiple testing with the Benjamin-Hochberg method. A random forest classifier was trained on 200 samples (the first cohort) and tested on 80 samples (the second cohort) using the random forest package in R. Correlations between enriched species and circulating metabolites were tested with Spearman's correlation. In the mouse study, the sample distribution was determined using the Kolmogorov–Smirnov normality test. For parametric data, a two-tailed unpaired Student's t test was used to analyze the differences between two groups. One-way ANOVA with the Bonferroni post hoc test was performed when more than 2 groups were evaluated. For nonparametric data, the Mann-Whitney U test with the exact method was used to analyze the differences between two groups. The rates of aneurysm formation and rupture in each group were evaluated using Fisher's exact test. The cumulative incidence of signs of stroke was evaluated using a log-rank test. P < 0.05 was considered statistically significant.

**Reporting summary**. Further information on research design is available in the Nature Research Reporting Summary linked to this article.

## Data availability
All relevant data are available from the authors. The RNA sequencing data and the metagenomics data have been deposited in National Genomics Data Center, Beijing Institute of Genomics (BIG), Chinese Academy of Sciences, under accession number PRJCA001337. Furthermore, the metagenomic sequencing data from the 63 fecal samples from the controls in the first cohort have been previously published (IDs for each sample are shown in Supplementary Data 1), and are available through the EMBL European Nucleotide Archive (ENA) under BioProject accession code PRJEB13870 [https://www.ebi.ac.uk/ena/browser/view/PRJEB13870]. Targeted metabolomics analysis are provided in files named Supplementary Data 11 and 17. The source data underlying Figs. 1b, c, e, f, 2, 4a, 5d, 6d–h, 8a, b, f–j and Supplementary Figs. 3, 7, 8b, 11–13, 15, 17a are provided as a Source Data file. Source data are provided with this paper.

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

## Acknowledgements

The authors thank M. Fang and L. Zhu for their assistance in cultivating bacteria. We also thank B. Geng and C. Jiang for reviewing the manuscript. This work was supported by the National Basic Research Program of China (2014CB541601 to J.C.), the Chinese Academy of Medical Sciences Innovation Fund for Medical Sciences (2016-I2M-1-015 to J.C.), the National Natural Science Foundation of China (91539113 to J.C. and 81800263 to H.L.), and Fundamental Research Funds for the Central Universities (3332018050 to H.L.).

## Author contributions

H.L. and H.X. performed the experiments and analyzed and interpreted the data. Y.L., Y.J., Y.H., T.L., X.T., X.Z., Y.Z., S.W., C.Z., J.G., X.W., H.W., C.B., Y.S., L.S., Y.Z., R.H., and J.C. performed the experiments. H.L., H.X., and J.C. designed the study and wrote the manuscript.

## Competing interests

The authors declare no competing interests.
