## [Peer Review File · Nature Communications]

Reviewers' Comments:

Reviewer #1:

Remarks to the Author:

A recent study has proposed an association between alterations in gut microbial composition and the incidence of intracranial aneurysms (IAs). This is analogous to studies showing an association between alterations in gut microbial composition and the incidence of intracranial cavernous malformations. In the present study, the authors performed a metagenome-wide association study between 140 Chinese patients with UIAs and 140 controls. The authors note 17 species, including *Parabacteroides* sp. and *Odoribacter splanchnicus*, revealing microbial changes in the deteriorating gut microbiota of UIA patients. They identified UIA-associated species linked to changes in circulating amino acids, including taurine. By fecal transplantation from UIA patients and control donors into mice, they note that UIA gut microbiota contributed to the formation and rupture of intracranial aneurysms. Taurine supplementation partly reversed the progression of intracranial aneurysms.

A recent study reported that depletion of the gut microbiota by antibiotics reduces the incidence of intracranial aneurysms in mice (1). Nevertheless, direct evidence for altered gut microbial composition in UIA patients and the precise underlying mechanisms are still lacking.

How did the authors come to the total number of human patients and mice? Were power analysis done before experiments?

In the human cohort, total of 140 UIA patients and 140 age-, sex- and blood pressure-matched controls were collected from 2 cohorts. Were other factors similarly distributed between cases and controls such as smoking (2)? I believe they said that none of the patients had a familial history of IA?

Interesting that a prior study report that gut bacteria may transmigrate to cerebral arteries and play a direct role in the pathogenesis of UIA, independent of a metabolite-based mechanism (3). However, the current authors did not detect bacterial DNA in the cerebral arteries in UIA patients or mice with UIAs.

Why did the authors ultimately focus on Taurine? They performed some many experiments with sequencing in humans and mice, but did not use this other than to show differential regulation. Ultimately they also performed metabolomics profiling of fatty acids and amino acids in mice after fecal transplantation. In total, the serum concentrations of 2 of 8 fatty acids and 8 of 38 amino acids differed substantially between mice transplanted with feces from UIA patients and controls (Supplementary Table 12). Specifically, among these altered metabolites, taurine, L-histidine, and linoleic acid also corroborated the lower abundance in UIA patients than in controls. Did the authors assess L-histidine or LA for their effects on aneurysms? Did they assess alterations in taurine in human UIA patients. This would make for a much stronger argument and the potential that taurine supplementation may have an impact in human UIA patients.

Is there any way to downregulate taurine to see if it leads to an increase incidence of aneurysm formation, progression, rupture?

The authors found that taurine altered macrophages, neurophils, TNF, IL, MMPs. These have all been shown to have an impact on aneurysm formation and rupture in vivo via genetic inactivation or pharmacological inhibition. TNF and MMPs have also been found to be upregulated in human aneurysms. These sources could be further referenced to provide support for their mechanistic studies (2, 4-9).

At an elastase dose of 35 mU no mice died either following surgery or from aneurysm rupture?

I must have missed it somewhere in the supplement but when were samples collected in patients with UIA in relation to treatment of their UIAs? How were controls selected?

REFERENCES

1. Shikata, F. et al. Potential Influences of Gut Microbiota on the Formation of 13 Intracranial Aneurysm. *Hypertension* 73, 491-496 (2019).
2. Starke, RM et al. Cigarette Smoke Initiates Oxidative Stress-Induced Cellular Phenotypic Modulation Leading to Cerebral Aneurysm Pathogenesis. *Arterioscler Thromb Vasc Biol.* 2018 Mar;38(3):610-621.
3. Pyysalo, M. J. et al. Bacterial DNA findings in ruptured and unruptured 30 intracranial aneurysms. *Acta Odontol. Scand.* 74, 315-320 (2016).
4. Starke RM et al. Critical role of TNF- α in cerebral aneurysm formation and progression to rupture. *J Neuroinflammation.* 2014 Apr 16;11:77
5. Kanematsu. Critical roles of macrophages in the formation of intracranial aneurysm. *Stroke.* 2011 Jan;42(1):173-8
6. Ali et al. TNF- α induces phenotypic modulation in cerebral vascular smooth muscle cells: implications for cerebral aneurysm pathology. *J Cereb Blood Flow Metab.* 2013 Oct;33(10):1564-73.
7. Hasan et al. Macrophage imbalance (M1 vs. M2) and upregulation of mast cells in wall of ruptured human cerebral aneurysms: preliminary results. *J Neuroinflammation.* 2012 Sep 21;9:222.
8. Jayaraman et al. TNF-alpha-mediated inflammation in cerebral aneurysms: a potential link to growth and rupture. *Vasc Health Risk Manag.* 2008;4(4):805-17.
9. Kim et al. Matrix metalloproteinase-9 in cerebral aneurysms. *Neurosurgery.* 1997 Sep;41(3):642-66;

Reviewer #2:

Remarks to the Author:

Li et al. performed metagenome-wide association analysis on unruptured intracranial aneurysms (UIA), demonstrated causal role of the gut microbiome by transplanting UIA feces to mice and also shown that Taurine supplementation could partly reverse the progression of UIA. This is a comprehensive study, as authors have tried to: 1) identify microbial difference between UIA and controls; 2) establish the link between UIA microbes to circulating levels of fatty acids and amino acids; 3) prove causality via faecal transplantation; 4) investigate the impact of UIA microbiome on transcriptional profile of cerebral vessels; and 5) the impact of Taurine supplement on the progression of UIA. The findings are relevant to the field. However, I have several concerns:

The authors qualified and quantified taxa and KEGG pathways based on MLGs (metagenomic linkage groups). The biological interpretation of these MLGs is difficult as they are based on co-abundance clustering. Nowadays, MetaPhlan and HUMAnN2 are two commonly used tools for taxa and functional assignments based on metagenomics sequencing data. Why did authors still choose to use MLG? Author should also compare their taxa and pathway assignments to the results of

MetaPhlan and HUMAnN2.

Authors did not observe significant difference in α -diversity at the genus level. How about the α -diversity at the species level?

The fecal transplantation experiment is nice as it demonstrated the impact of gut microbiome on IA. However, it is not clear how these donors (2 UIA and 2 controls) were selected? To what extent they represented the identified UIA microbial composition. The comparison of 4 genera in Sup Fig 6 is not informative. Why were only 4 genera detected? To my surprise, the genera abundances between two UIA donors were very different. The same was observed between two control donors. For instance: the abundance of Bacteroides in UIA2 donor (80%) was nearly 2 times higher than that in UIA1 donor (40%). Were mice single-housed? The authors need to add the link between donors and receivers and perform statistical analysis to demonstrate that mice receivers did have more similar microbial profiles as the human donors.

Which microbial difference in fecal transplanted mice could explain the observed difference in IA phenotype? Were they in line with the associated microbial features identified in metagenome association in human?

It is difficult to interpret the co-abundance network analysis. It is unclear how this part can aid a better understanding on the microbial difference between UIAs and controls. It is unclear how the spearman correlation coefficient 0.6 was chosen for network construction. Moreover, it is well known that spearman correlation is not an ideal method for co-abundance analysis on compositional data. Several tools, like SparCC has been proposed for this purpose. Comparison of Spearman to other network construction methods have been extensively assessed in <https://www.ncbi.nlm.nih.gov/pmc/articles/PMC4918442/>

Page 8-9 show patients can be classified based on 17 MLS. The cross-validation did not make any sense. These 17 MLS were selected as their associations were shared in both cohorts. Thus both cohorts have been used for discovery. The cross-validation using the same (or sub) cohorts cannot be considered as independent validation. There is an overfitting problem.

The authors did not fill in the Report Summary accurately. For instance, graphic software (Graphpad, ImageJ and Adobe illustrators) was mentioned as data analysis software.

Fig 1: Authors should rarefy the number of reads to assess the impact of sequencing depth.

Minor:

Sup Fig 9: L-histidine association in human and mice were not consistent. Were colors of mouse experiment swapped in figure?

Page 6: subtitle "microbial strains associated with UIAs". There is no strain analysis.

Page 10: we investigated the microbe-host interactions in UIA patients. There was no interaction analysis.

Page 22: patients and controls were enrolled through different hospitals and from different areas. Authors should also discuss other potential confounding effects. It is not true that Han population share similar dietary habits (page 23).

Authors should provide accurate P values for all statistical analyses.

Reviewer #3:

Remarks to the Author:

The authors present a novel body of work examining the fecal microbiome of individuals with intracranial aneurysms. Major strengths of the study include the second cohorts that validate findings, corroborating metabolomic analyses, an in vivo murine model demonstrating causality, and furthermore identification of a potentially translatable strategy that mediates a meaningful benefit in the pre-clinical model.

From the methods, it seems that some of the control patients were enrolled at different sites from the cases - could this be a potential confounder?

Figure 1E - PCA indicates that the authors utilized a Euclidean distance matrix to generate these plots. Typically, microbiome studies will instead utilize a Bray-curtis dissimilarity (already calculated in 1D) which is more suited for abundance data, or even more commonly a weighted UniFrac distance which incorporates abundance and phylogenetic information. The authors can also quantify if the samples cluster separately with statistical significance; commonly used tools included ANOSIM and ADONIS ("permutational manova"), which are available in the vegan R package.

Figure 1F - are any of these significant?

SFig 6 - a known limitation of human-into-mouse FMT experiments is that large prominent groups of human bacteria fail to take in mice, in particular Clostridia. Can the authors more clearly convey which bacteria did and did not take? SFig6A indicates that PC1 clearly distinguishes human samples from mouse samples. At what time point were mouse samples assayed? Did the authors monitor for stability of take of human bacteria? How much of the microbiome in these mice were residual mouse flora that returned after antibiotics?

Are the authors able to identify which bacteria in the control subjects, and in the mice, are responsible for taurine production? Can they show that these bacteria are differentially abundance in the human cohorts and in the mice?

Did the authors assay to see if their taurine intervention in the mice normalized serum levels?

The authors could consider citing a recent study where taurine was found to be reduced in mice receiving FMT from autistic individuals, and taurine supplementation improved autistic-like behavior in mice. Is autism known to be associated with intracranial aneurysms?

The following is the point-to-point response to the Reviewers' comments.

Reviewer #1 (Remarks to the Author):

A recent study has proposed an association between alterations in gut microbial composition and the incidence of intracranial aneurysms (IAs). This is analogous to studies showing an association between alterations in gut microbial composition and the incidence of intracranial cavernous malformations. In the present study, the authors performed a metagenome-wide association study between 140 Chinese patients with UIAs and 140 controls. The authors note 17 species, including Parabacteroides sp. and Odoribacter splanchnicus, revealing microbial changes in the deteriorating gut microbiota of UIA patients. They identified UIA-associated species linked to changes in circulating amino acids, including taurine. By fecal transplantation from UIA patients and control donors into mice, they note that UIA gut microbiota contributed to the formation and rupture of intracranial aneurysms. Taurine supplementation partly reversed the progression of intracranial aneurysms.

A recent study reported that depletion of the gut microbiota by antibiotics reduces the incidence of intracranial aneurysms in mice (1). Nevertheless, direct evidence for altered gut microbial composition in UIA patients and the precise underlying mechanisms are still lacking.

1. How did the authors come to the total number of human patients and mice? Were power analysis done before experiments?

Response: Thanks for pointing out this issue. For the clinical study, the original sample size for UIA patients and their paired controls were not determined by specific statistical hypothesis. However, from the aspect of post-hoc power analysis, the 100 pairs of participants for testing and the 40 pairs for validation can be interpreted as follows. The main purpose of the case-control studies was to detect and confirm the potential changes in gut microbiota, so, with a 2-sided 0.05 alpha level, 100 pairs of subjects would offer >80% power to detect a 0.4 standardized difference between IUA patients and controls. In the validation set, the sample size was 40:40 (for UIA and

control respectively). This number of participants also has ~80% power to demonstrate a minimum of 0.6 standardized difference between the paired groups at the 2-sided 0.05 significance level (to confirm the observed difference from training set). Multiplicity adjustment was not integrated into above calculations, because the findings on microbiota changes would be validated by further animal experiments.

For the animal studies, separate interventional studies were done with regard to different evaluation objectives. The original size of each animal study was determined by the experience of experts in the relevant research area. We also conducted post-hoc power analysis to show the detection ability (the minimum between group standardized difference) under the range of sample sizes used in our animal studies. The calculation did not bind to any specific primary endpoint (just to show the detection ability of the statistical test) and the results are summarized below.

Size of the study	Alpha level	Power	Standardized difference detectable
n = 5	2-sided 0.05	80%	1.78
n = 10	2-sided 0.05	80%	1.26
n = 15	2-sided 0.05	80%	1.03
n = 20	2-sided 0.05	80%	0.89

2. In the human cohort, total of 140 UIA patients and 140 age-, sex- and blood pressure-matched controls were collected from 2 cohorts. Were other factors similarly distributed between cases and controls such as smoking (2)? I believe they said that none of the patients had a familial history of IA?

Response: As shown in revised Supplementary Tables 19 and 20, the baseline characteristics of patients with UIA and controls, including cigarette smoking, were similarly distributed and were not significantly different in the two cohorts.

Actually, a family history of intracranial aneurysm was one of the exclusion criteria for the UIA patients. None of the 140 UIA patients had a familial history. We apologize for the confusion and have added clarifying statements to the revised

manuscript (See page 25, line 15).

3. Interesting that a prior study report that gut bacteria may transmigrate to cerebral arteries and play a direct role in the pathogenesis of UIA, independent of a metabolite-based mechanism (3). However, the current authors did not detect bacterial DNA in the cerebral arteries in UIA patients or mice with UIAs.

Response: Thanks for pointing out this issue. Whether gut bacteria can transmigrate to cerebral arteries and play a direct role in the pathogenesis of intracranial aneurysm remains controversial. Pyysalo *et al.* demonstrated the presence of DNA from oral pathogens in the wall of aneurysms from Finnish patients ¹. Contrary to this study, a recent study has shown that no oral or other bacteria can be found in the wall of intracranial aneurysms in a French population ².

In the present study, we did not obtain specimens from unruptured aneurysm walls during the surgical clipping because the procedure has potential risk for the patients, such as bleeding. However, as suggested by this reviewer, we further searched for bacterial DNA in the cerebral arteries of mice. Cerebral arteries (anterior and posterior cerebral arteries, anterior and posterior communicating arteries, middle cerebral arteries, and basilar arteries) in each group were isolated for quantitative real-time PCR. The cerebral arteries from the sham group served as a negative control. We detected no bacterial DNA in the cerebral arteries of mice transplanted with feces either from UIA patients or from controls. This absence of bacterial infection of the cerebral arteries highlights the taurine-based mechanism in the pathogenesis of UIA. However, more investigations are required to establish the precise role of bacterial infection in the progression of unruptured intracranial aneurysms.

Assessment of bacterial DNA in the cerebral arteries of mice that received fecal microbiota transplantation (FMT). Quantitative real-time PCR was used to evaluate the expression of total bacteria of the cerebral arteries of mice. Data are the mean \pm SD. n = 5. One-way ANOVA with the Bonferroni post hoc test.

Sequence of each primer used in real-time PCR

Gene Symbol	Sequence
Total bacteria	Forward: 5'-ACTCCTACGGGAGGCAGCAG-3' Reverse: 5'-ATTACCGCGGCTGCTGG-3'
GAPDH	Forward: 5'-CTCATGACCACAGTCCATGC-3' Reverse: 5'-CACATTGGGGGTAGGAACAC-3'

4. Why did the authors ultimately focus on Tuarine? They performed some many experiments with sequencing in humans and mice, but did not use this other than to show differential regulation. Ultimately they also performed metabolomics profiling of fatty acids and amino acids in mice after fecal transplantation. In total, the serum concentrations of 2 of 8 fatty acids and 8 of 38 amino acids differed substantially between mice transplanted with feces from UIA patients and controls (Supplementary Table 12). Specifically, among these altered metabolites, taurine,

L-histidine, and linoleic acid also corroborated the lower abundance in UIA patients than in controls. Did the authors assess L-histidine or LA for their effects on aneurysms?

Response: Sorry for the confusion. In the present study, we demonstrated that impairment of the UIA microbiome evoked a disease-linked state through interference with physiological metabolic functions, especially fatty-acid and amino-acid metabolism. Therefore, we further functionally characterized the UIA microbiome based on the metabolism of fatty-acids and amino-acids. In total, the serum concentrations of 7 of 9 fatty acids and 12 of 32 amino acids (**including taurine, see revised Supplementary Table 11**) differed substantially between human UIA patients and controls. Furthermore, after fecal transplantation, among the 10 altered circulating metabolites between mice transplanted with feces from UIA patient and mice transplanted with control feces, only 3 metabolites (taurine, L-histidine, and linoleic acid) were also substantially different in human UIA patients than in controls. On this basis, we next determined whether supplementation with taurine, L-histidine, or linoleic acid has potential therapeutic value for reducing the formation and rupture of intracranial aneurysms in mice. We found that supplementation with taurine, but not L-histidine or linoleic acid, reduced the formation and rupture of intracranial aneurysms in mice (**see revised Fig. 6 and Supplementary Fig. 16**).

Furthermore, we found that the abundance of *Hungatella hathewayi* was markedly decreased in the UIA patients group no matter whether it was based on MLGs (metagenomic linkage groups) or on MetaPhlan (**see revised Fig. 2 and Supplementary Fig. 7**). An enriched abundance of *H. hathewayi* was also found in mice treated with control feces, compared with that in mice treated with UIA patient feces (**see revised Fig. 7b**). More importantly, *H. hathewayi* was positively correlated with serum taurine concentration in humans and in mice after fecal transplantation (**see revised Fig. 4b and Fig. 7a**). To explore the possible causal role of *H. hathewayi* in the regulation of the level of circulating taurine and the formation and rupture of intracranial aneurysms, mice were gavaged with live *H. hathewayi* or vehicle after fecal transplantation. The plasma concentration of taurine was significantly restored

after *H. hathewayi* administration. Of note, *H. hathewayi* protected mice against the formation and rupture of intracranial aneurysms (see revised Fig. 8), clearly demonstrating that *H. hathewayi* in the gut plays a pivotal role in controlling taurine metabolism and the progression of intracranial aneurysms in the host.

Did they assess alterations in taurine in human UIA patients. This would make for a much stronger argument and the potential that taurine supplementation may have an impact in human UIA patients.

Response: Yes, we measured the taurine levels in UIA patients and controls. We found that the serum concentrations of taurine were significantly lower in UIA patients than in controls (see revised Supplementary Fig. 12 and Supplementary Table 11).

5. Is there any way to downregulate taurine to see if it leads to an increase incidence of aneurysm formation, progression, rupture?

Response: Thanks for pointing out this critical issue. Recent studies have reported that oral administration of a 3% β -alanine solution in the drinking water is as effective in increasing urinary taurine excretion and decreasing circulating taurine levels in rodent as subcutaneous or intraperitoneal administration³⁻⁵. Accordingly, in the present study, after depletion of the gut microbiota by antibiotic cocktail, mice were transplanted with feces from control donors. Then mice were supplemented with 3% β -alanine in the drinking water for 2 weeks, which significantly decreased the plasma concentration of taurine. Then the mice underwent aneurysm-induction surgery and continued to drink β -alanine or Vehicle until they developed neurological symptoms or until day 21 after surgery if the mouse did not have neurologic signs. Compared with Vehicle treatment, β -alanine treatment significantly increased the overall incidence of aneurysms and the rupture rate. Kaplan–Meier analysis further revealed a significant increase in aneurysmal rupture with β -alanine administration. Taken together, these data clearly demonstrated that the diminished level of taurine increases the incidence of the formation and rupture of intracranial aneurysms in mice.

Diminished level of circulating taurine increases the formation and rupture of intracranial aneurysms in mice. (a) Serum concentrations of taurine after β -alanine or Vehicle treatment measured by UHPLC-MS/MS ($n = 10$; Student's unpaired two-tailed t-test). (b) Representative images of intracranial aneurysms induced with angiotensin II and elastase in each group. (c) Incidence of unruptured and ruptured aneurysms 21 days after aneurysm induction ($n = 20$). (d) Cumulative symptom-free curves for mice with aneurysms to show the time course of symptom onset ($n = 10$ – 17 ; log-rank (Mantel–Cox) test).

6. The authors found that taurine altered macrophages, neutrophils, TNF, IL, MMPs. These have all been shown to have an impact on aneurysm formation and rupture in vivo via genetic inactivation or pharmacological inhibition. TNF and MMPs have also been found to be upregulated in human aneurysms. These sources could be further referenced to provide support for their mechanistic studies (2, 4-9).

Response: Thanks for your helpful suggestion. We have added these references to the revised manuscript.

7. At an elastase dose of 35 mU no mice died either following surgery or from aneurysm rupture?

Response: We apologize for the confusion. In the present study, a Hamilton syringe with a 26G needle was advanced into the right basal cistern of mice. Elastase solution (35 mU) was then injected. After elastase injection and closure of the skin over the burr hole, a separate small incision was made in the dorsal skin between the scapulae. A micro-osmotic pump containing angiotensin II was implanted into a subcutaneous pocket. There was no mortality during this surgical procedure. We have revised the statement in the revised manuscript (see page 13, lines 19–20).

8. I must have missed it somewhere in the supplement but when were samples collected in patients with UIA in relation to treatment of their UIAs? How were controls selected?

Response: We apologize for the confusion. The samples from UIA patients were collected prior to the microsurgical clipping. Patients who were treated for their UIAs before sample collection were excluded. We added this to the revised manuscript (see page 25, lines 18–19).

In both cohorts, controls were outpatients with minor illnesses and matched to UIA patients for age, sex, and blood pressure. Controls were free of aneurysmal symptoms or a history of subarachnoid hemorrhage following the same exclusion criteria as UIA patients. We added this to the revised manuscript (see page 25, lines 21-22 and page 26, lines 1-2).

Reviewer #2 (Remarks to the Author):

Li et al. performed metagenome-wide association analysis on unruptured intracranial aneurysms (UIA), demonstrated causal role of the gut microbiome by transplanting UIA feces to mice and also shown that Taurine supplementation could partly reverse the progression of UIA. This is a comprehensive study, as

authors have tried to: 1) identify microbial difference between UIA and controls; 2) establish the link between UIA microbes to circulating levels of fatty acids and amino acids; 3) prove causality via faecal transplantation; 4) investigate the impact of UIA microbiome on transcriptional profile of cerebral vessels; and 5) the impact of Taurine supplement on the progression of UIA. The findings are relevant to the field. However, I have several concerns:

1. The authors qualified and quantified taxa and KEGG pathways based on MLGs (metagenomic linkage groups). The biological interpretation of these MLGs is difficult as they are based on co-abundance clustering. Nowadays, MetaPhlan and HUMAnN2 are two commonly used tools for taxa and functional assignments based on metagenomics sequencing data. Why did authors still choose to use MLG? Author should also compare their taxa and pathway assignments to the results of MetaPhlan and HUMAnN2.

Response: Thanks for pointing out this critical issue. We agree with the reviewer and have done the analyses using MetaPhlan2 and HUMAnN2 as suggested.

MetaPhlan2 was run to identify the taxonomic abundances at the species level. In the first cohort, a total of 47 species differed significantly in abundance between UIA and control samples, 38 of them being more abundant in UIA samples (see **revised Supplementary Table 5**). Among these 47 species, 8 (17.0%) were also differentially enriched in UIA patients compared with controls in the second cohort (see **revised Fig. 2 and Supplementary Table 6**). Notably, among the 17 MLG-associated species and 8 MetaPhlan2-associated species that differed in abundance between the two cohorts, we found that *H. hathewayi*, *O. splanchnicus* and *Alistipes putredinis* were considerably overlapped species; this revealed the most representative microbial changes in the deteriorating gut microbiota of UIA patients and are more likely to be involved in the progression of UIA.

We next looked to characterize the species-level functional profiling in UIA using the HUMAnN2 pipeline. We identified a total of 46 metabolic pathways that were differentially abundant between UIA patients and controls (see **revised Fig. 4a and**

Supplementary Table 9). Within these 46 pathways, those contributing to amino acid and fatty acid metabolism were adequately represented. Specifically, the microbiome of UIA patients was found to be significantly dominated by unsaturated fatty acid biosynthesis. Moreover, the biosynthesis of the amino acids threonine, isoleucine, lysine, and methionine dominated the metabolic landscape of the microbiome in controls. The rest of the pathways were mainly glucose and nucleotide metabolic pathways. We also compared those pathways that differed in abundance between UIA patients and controls to the 25 KEGG pathways based on MLGs that also differed in abundance between UIA patients and controls. Notably, unsaturated fatty acid biosynthesis, amino acid (methionine, tryptophan, pyruvate, and isoleucine) metabolism, pyruvate metabolism, and glycolysis were overlapped microbial pathways that were differentially abundant using both HUMAnN2 and KEGG.

2. Authors did not observe significant difference in α -diversity at the genus level.

How about the α -diversity at the species level?

Response: We apologize for the confusion. The α -diversity at the species level was calculated as suggested by the reviewer. We also did not find a significant difference in α -diversity at the species level (see revised **Supplementary Fig. 3**).

3. The fecal transplantation experiment is nice as it demonstrated the impact of gut microbiome on IA. However, it is not clear how these donors (2 UIA and 2 controls) were selected? To what extent they represented the identified UIA microbial composition. The comparison of 4 genera in Sup Fig 6 is not informative. Why were only 4 genera detected? To my surprise, the genera abundances between two UIA donors were very different. The same was observed between two control donors. For instance: the abundance of *Bacteroides* in UIA2 donor (80%) was nearly 2 times higher than that in UIA1 donor (40%). Were mice single-housed? The authors need to add the link between donors and receivers and perform statistical analysis to demonstrate that mice receivers did have more similar microbial profiles as the human donors.

Response: Thanks for pointing out this critical issue and we apologize for the confusion. Deep shotgun sequencing and metagenome-wide association studies have enabled more in-depth characterization and insights into the function of gut microbiomes than 16S rDNA amplicon sequencing. Furthermore, the 16S-based sequencing approach probably lacks the required species-level resolution. On these bases, all bacterial DNA samples including the 2 UIA donors and 2 control donors and all recipient mice post-transplantation were sequenced on the Illumina HiSeq X Ten platform for metagenomic sequencing.

As expected, PCoA at the species level showed that the 2 control donors were distributed near the center of gravity of the 100 controls, and the 2 UIA donors were distributed near the center of gravity of the 100 UIA patients, which indicated that the human donors represented the identified features of the respective gut microbiota (**Supplementary Fig. 8a**). Mice were single-housed throughout the experiments. After fecal transplantation, among the top 20 most abundant genera in humans, ten genera (labeled in red in the figure) including *Bacteroides*, *Prevotella*, and *Alistipes* were still abundant in mice treated with donor feces.

Furthermore, to demonstrate to what extent mouse recipients have microbial profiles similar to the human donors, we performed SourceTracker analyses to determine the extent of donor engraftment using a Bayesian algorithm^{6,7}. We found that 61.5% of the fecal bacterial communities in mice treated with control feces were attributable to the control donors. Consistently, 53.8% of the fecal bacterial communities in mice treated with UIA feces were attributable to the UIA donors (**Supplementary Fig. 8b**). This overall degree of microbiota engraftment was similar

to a recent study ⁷, which indicated that mouse recipients have microbial profiles more similar to human donors.

More importantly, we found that the lower relative abundances of *H. hathewayi* were all found in UIA patients in the two cohorts, UIA donors, and mice treated with UIA patient feces than in their respective controls (see revised Fig. 7b). Oral gavage with *H. hathewayi* protected against the formation and rupture of intracranial aneurysms in mice (see revised Fig. 8).

4. Which microbial difference in fecal transplanted mice could explain the observed difference in IA phenotype? Were they in line with the associated microbial features identified in metagenome association in human?

Response: Thanks for pointing out this critical issue. In the present study, to validate the altered circulating metabolites in UIA patients and to further assess the relationships between gut microbiota and circulating metabolites, we performed targeted metabolomics profiling of fatty acids and amino acids in mice after fecal transplantation. Specifically, among these altered metabolites, taurine, L-histidine, and linoleic acid also corroborated the lower abundance in UIA patients than in controls (see revised Supplementary Fig. 12).

We next determined whether supplementation with taurine, L-histidine, or linoleic acid has potential therapeutic value for reducing the formation and rupture of intracranial aneurysms in mice. We found that supplementation with taurine, but not L-histidine or linoleic acid, reduced the formation and rupture of intracranial aneurysms in mice (see revised Fig. 6 and Supplementary Fig. 16).

To further investigate the extent to which the altered microbiome in UIAs is associated with the altered circulating taurine in the host, we calculated the Spearman's rank correlations between altered metabolites and altered species in mice after fecal transplantation. We found that 5 species were positively correlated and 1 species was negatively correlated with taurine levels (see revised Fig. 7a). Among these 6 species, only *H. hathewayi* was an overlapped species that also positively correlated with taurine levels in the human study. Furthermore, the lower relative

abundances of *H. hathewayi* all occurred in the UIA patients of the two cohorts, UIA donors, and mice treated with UIA patient feces, relative to their respective controls (see revised Fig. 7b).

On these bases, to explore the possible causal role of *H. hathewayi* in the regulation of the level of circulating taurine and the formation and rupture of intracranial aneurysms, mice were gavaged with live *H. hathewayi* or vehicle (sterile PBS) after fecal transplantation. We thus provide novel and direct evidence that *H. hathewayi* affects circulating taurine concentration and protects mice against the formation and rupture of intracranial aneurysms (see revised Fig. 8).

5. It is difficult to interpret the co-abundance network analysis. It is unclear how this part can aid a better understanding on the microbial difference between UIAs and controls. It is unclear how the spearman correlation coefficient 0.6 was chosen for network construction. Moreover, it is well known that spearman correlation is not an ideal method for co-abundance analysis on compositional data. Several tools, like SparCC has been proposed for this purpose. Comparison of Spearman to other network construction methods have been extensively assessed in <https://www.ncbi.nlm.nih.gov/pmc/articles/PMC4918442/>

Response: Thanks for the helpful suggestion. Through careful reading of the literature provided by the reviewers, we realized that Spearman correlation, which was used to construct the network structure of species with differential abundance, is indeed not an ideal method for co-abundance analysis of compositional data. Instead, the co-occurrence network based on sparse correlations of species with differential abundance was reconstructed using SparCC (Sparse Correlations for Compositional data).

In the present study, we clustered the genes that displayed significant differences in abundance between the two groups into MLGs. These MLGs were used to construct an MLG network depicting the correlation between UIA-associated gut microbial markers. Notably, UIA-enriched MLGs were more highly interconnected than control-enriched MLGs. We acknowledge that the co-abundance network is not

adequately informative and is limited in explaining the microbial difference between UIAs and controls. Thus, we moved this result to the supplemental material (see revised Supplementary Fig. 4).

6. Page 8-9 show patients can be classified based on 17 MLS. The cross-validation did not make any sense. These 17 MLS were selected as their associations were shared in both cohorts. Thus both cohorts have been used for discovery. The cross-validation using the same (or sub) cohorts cannot be considered as independent validation. There is an overfitting problem.

Response: We apologize for the overfitting problem. Because MetaPhlAn2 was run to identify the taxonomic abundances at the species level, we reconstructed a random forest classifier from the 200 UIA and control samples of the first cohort. Cross-validation and receiver operating characteristic curves for distinguishing UIA patients from controls were developed again. We were able to detect UIA patients accurately based on the 47 species that were differentially abundant identified by MetaPhlAn2 in the first cohort, as indicated by an area under the receiver operating curve (AUC) of up to 0.86 and a 95% confidence interval (CI) of 0.81–0.91 (see revised Fig. 3a). Consistent with these results, the classification error remained relatively low in the second cohort (80 UIA and control samples), with an AUC of 0.72 and a 95% CI of 0.65–0.79 (see revised Fig. 3b).

7. The authors did not fill in the Report Summary accurately. For instance, graphic software (Graphpad, ImageJ and Adobe illustrators) was mentioned as data analysis software.

Response: We apologize for the mistake. We have revised the Reporting Summary accordingly.

8. Fig 1: Authors should rarefy the number of reads to assess the impact of sequencing depth.

Response: Thanks for the helpful suggestion. The sequenced data from each sample

were aligned back to the assembled pan-genome. Then the sequence was randomly sampled at a gradient of 2M reads. The results showed that the number of unigenes reached saturation when the number of sequencing reads reached 30M (~4.5G bases). Most of samples we sequenced in the present study exceeded 4.5G of the data.

9. Sup Fig 9: L-histidine association in human and mice were not consistent. Were colors of mouse experiment swapped in figure?

Response: We apologize for the mistake. The red and blue labels were swapped in the figure of L-histidine concentrations in mice. We have modified it (see revised Supplementary Fig. 12).

10. Page 6: subtitle “microbial strains associated with UIAs”. There is no strain analysis.

Response: Sorry for the typo. We've corrected it to “Microbial species associated with UIAs”.

11. Page 10: we investigated the microbe-host interactions in UIA patients. There was no interaction analysis.

Response: We apologize for the mistake. We have deleted the statement.

12. Page 22: patients and controls were enrolled through different hospitals and

from different areas. Authors should also discuss other potential confounding effects. It is not true that Han population share similar dietary habits (page 23).

Response: Thanks for pointing out this issue. In the present study, some of the controls were indeed enrolled in Hebei Province (Cangzhou and Tangshan), while all UIA patients were enrolled in Beijing. First, Beijing is not so far from the above two cities in Hebei Province (about 200 km). Second, in the second cohort, 10 controls were consecutively enrolled from Cangzhou Central Hospital (Hebei Province) and 30 from Tsinghua University Hospital (Beijing). To further evaluate whether the regional factor is a potential confounder affecting the features of the gut microbiota, PCoA based on the features of the gut microbiota at the genus level was applied to discriminate control individuals from different areas. As expected, the PCoA showed a similar spatial distribution between the 2 groups, indicating that control individuals from different areas shared the similar features of gut microbiota and the region may not be a confounding factor in this study. We have added this to the revised manuscript (see revised **Supplementary Fig. 2**).

We apologize for the mistake and have deleted the statement “This study was conducted in North China among individuals from the Han population who share similar dietary habits.”

13. Authors should provide accurate *P* values for all statistical analyses.

Response: We agree with the reviewer and the accurate *P* values for all statistical analyses have been provided. In particular, the accurate *P* values for the statistical analyses of differential abundances of gut microbiota at the genus and species levels can be found in the respective Supplementary Tables.

Reviewer #3 (Remarks to the Author):

The authors present a novel body of work examining the fecal microbiome of individuals with intracranial aneurysms. Major strengths of the study include the second cohorts that validate findings, corroborating metabolomic analyses,

an in vivo murine model demonstrating causality, and furthermore identification of a potentially translatable strategy that mediates a meaningful benefit in the pre-clinical model.

1. From the methods, it seems that some of the control patients were enrolled at different sites from the cases - could this be a potential confounder?

Response: Thanks for pointing out this critical issue, which was also raised by reviewer #2. In the present study, some of the controls were indeed enrolled in Hebei Province (Cangzhou and Tangshan), while all UIA patients were enrolled in Beijing. First, Beijing is not so far from the above two cities in Hebei Province (about 200 km). Second, in the second cohort, 10 controls were consecutively enrolled from Cangzhou Central Hospital (Hebei Province) and 30 from Tsinghua University Hospital (Beijing). To further evaluate whether the regional factor is a potential confounder affecting the features of the gut microbiota, PCoA based on the features of the gut microbiota at the genus level was applied to discriminate control individuals from different areas. As expected, the PCoA showed a similar spatial distribution between the 2 groups, indicating that control individuals from different areas shared the similar features of gut microbiota and the region may not be a confounding factor in this study. We have added this to the revised manuscript (see revised **Supplementary Fig. 2**).

2. Figure 1E - PCA indicates that the authors utilized a Euclidean distance matrix to generate these plots. Typically, microbiome studies will instead utilize a Bray-curtis dissimilarity (already calculated in 1D) which is more suited for abundance data, or even more commonly a weighted UniFrac distance which incorporates abundance and phylogenetic information. The authors can also quantify if the samples cluster separately with statistical significance; commonly used tools included ANOSIM and ADONIS ("permutational manova"), which are available in the vegan R package.

Response: We apologize for the confusion and thank this reviewer for the helpful

suggestions. For figure 1E, our initial purpose was to show the spatial distribution difference between UIA and the control group, rather than to show the phylogenetic clustering information between samples. We agree with the reviewer that using Bray-Curtis dissimilarity is more suitable for abundance data in the present study. According to the suggestions of this reviewer, we calculated Bray-Curtis dissimilarities for ordination by principal coordinate analysis (PCoA) and found that ordination of Bray-Curtis dissimilarity by PCoA revealed separation between the two groups. This separation was due to differences in community composition, evaluated by ANOSIM in the vegan R package ($P = 0.001$; see revised Fig. 1d and e).

3. Figure 1F - are any of these significant?

Response: We apologize for the confusion. The genera shown in Fig. 1F were all significantly different in abundance between the 2 groups. The accurate P values for the statistical analyses of all differential abundances of gut microbiota at the genus level can be found in the **revised Supplementary Tables 4**.

4. SFig 6 - a known limitation of human-into-mouse FMT experiments is that large prominent groups of human bacteria fail to take in mice, in particular Clostridia. Can the authors more clearly convey which bacteria did and did not take? SFig6A indicates that PC1 clearly distinguishes human samples from mouse samples. At what time point were mouse samples assayed? Did the authors monitor for stability of take of human bacteria? How much of the microbiome in these mice were residual mouse flora that returned after antibiotics?

Response: Thanks for pointing out this critical issue. We agree with the reviewer that large prominent groups of human bacteria fail to take in mice after fecal transplantation. In the present study, among the top 20 most abundant genera in human individuals, ten genera (labeled in red in the figure) including *Bacteroides*, *Prevotella*, and *Alistipes* were still abundant in mice treated with donor feces.

We also noted that *Clostridium*, which was abundant in human individuals, was indeed less abundant in mice after fecal transplantation. However, the abundance of a species named *Clostridium hathewayi*, which was reclassified as *Hungatella hathewayi* in 2014⁸, was successfully conveyed by fecal transplantation. Interestingly, our result is consistent with those of a previous study that found that the *H. hathewayi* was isolated from germ-free mice inoculated with a human fecal microbiota⁹. Moreover, we found that the lower relative abundances of *H. hathewayi* all occurred in UIA patients in the two cohorts, UIA donors, and mice treated with UIA patient feces compared with their respective controls (see revised Fig. 7b).

Fecal samples were collected from mice one week after fecal transplantation. To demonstrate to what extent mice recipients have microbial profiles similar to the human donors, we used SourceTracker analyses to determine the extent of donor engraftment using a Bayesian algorithm^{6,7}. We found that 61.5% of the fecal bacterial communities in mice treated with control feces were attributable to the control donors. Consistently, 53.8% of the fecal bacterial communities in mice treated with UIA feces were attributable to the UIA donors (Supplementary Fig. 8b). This overall degree of microbiota engraftment was similar to a recent study⁷.

To monitor for the stability of engraftment of human bacteria, we collected fecal samples from the same recipient mice 5 days after aneurysm-induction surgery and sequenced them on the Illumina HiSeq X Ten platform for metagenomic sequencing. Among the 10 genera that were abundant in both human donors and mice treated with donor feces, *Bacteroides*, *Parabacteroides*, *Alistipes*, *Escherichia*, *Lachnoclostridium*, *Phascolarctobacterium* and *Roseburia* (labeled in red in the figure) were still

comparably abundant after aneurysm-induction surgery, indicating a sustained engraftment of human microbiota into mice following fecal transplantation.

Depletion of the gut microbiota was achieved by administering a mouse broad-spectrum antibiotic cocktail. We found that the 4 genera *Akkermansia*, *Bacteroides*, *Klebsiella*, and *Escherichia* were not fully depleted by the antibiotic cocktail. After fecal transplantation and aneurysm-induction surgery, *Akkermansia* and *Bacteroides* were significantly more abundant, so we speculate that these 2 genera were residual mouse flora that returned after antibiotics.

5. Are the authors able to identify which bacteria in the control subjects, and in the mice, are responsible for taurine production? Can they show that these bacteria are differentially abundance in the human cohorts and in the mice?

Response: Thanks for pointing out this critical issue. In the revised manuscript, all bacterial DNA samples including the 2 UIA donors, 2 control donors, and all recipient

mice post-transplantation were sequenced on the Illumina HiSeq X Ten platform for metagenomic sequencing. To identify the microbial species after fecal transplantation, MetaPhlan2 was run to identify the taxonomic abundances at the species level in mice. There were 14 differentially abundant species between mice treated with UIA patient feces and mice treated with control feces (**see revised Supplementary Tables 15**).

To further investigate the extent to which the altered microbiome in UIAs is associated with the altered circulating taurine in the host, we calculated the Spearman's rank correlation between altered metabolites and altered species in mice after fecal transplantation. We found that 5 species were positively correlated and 1 species was negatively correlated with taurine levels (**see revised Fig. 7a**). Among these 6 species, *H. hathewayi* was the only overlapped species that was also positively correlated with taurine levels in the human study. Furthermore, the lower relative abundances of *H. hathewayi* all occurred in UIA patients of the two cohorts, UIA donors, and mice treated with UIA patient feces, relative to their respective controls (**see revised Fig. 7b**).

On these bases, to explore the possible causal role of *H. hathewayi* in the regulation of the level of circulating taurine and the formation and rupture of intracranial aneurysms, mice were gavaged with live *H. hathewayi* or vehicle (sterile PBS) after fecal transplantation. Importantly, the plasma concentration of taurine was also significantly increased after *H. hathewayi* administration. Together, we provide novel and direct evidence that *H. hathewayi* affects circulating taurine concentration and protects mice against the formation and rupture of intracranial aneurysms (**see revised Fig. 8**).

6. Did the authors assay to see if their taurine intervention in the mice normalized serum levels?

Response: Yes. Compared with UIA fecal treatment, taurine supplementation normalized the serum levels of taurine in mice. We have added these data to the revised manuscript (**see revised Supplementary Fig. 13**).

7. The authors could consider citing a recent study where taurine was found to be reduced in mice receiving FMT from autistic individuals, and taurine supplementation improved autistic-like behavior in mice. Is autism known to be associated with intracranial aneurysms?

Response: We would like to express our sincere thanks to this reviewer for providing us with the valuable literature. To our knowledge, there is no definite evidence that intracranial aneurysms are associated with autism spectrum disorder (ASD). Indeed, our results are analogous to this study showing that the circulating concentration of taurine is reduced in germ-free mice receiving fecal transplantation from individuals with ASD, and taurine supplementation improves repetitive and social behaviors in mice¹⁰. Nevertheless, the microbiome features of ASD described in this study are distinct from those of UIAs in the present study. Specifically, at the genus level, *Bacteroides* and *Parabacteroides* were absent from ASD samples, but these were abundant in UIA samples. Conversely, *Sutterella* was decreased in ASD, but increased in UIA. At the species level, *B. ovatus* and *P. merdae*, which were absent from most or all ASD samples, were abundant in UIA samples. We have cited this recent study and discussed it in the revised manuscript (see page 22, lines 2-7).

We believe that our revised manuscript is significantly strengthened. We hope that the extensive revision, will remove all concerns raised by the reviewers and lead to the ultimate acceptance of our manuscript for publication in *Nature Communications*.

Thank you very much for handling the review of our manuscript.

Sincerely,

Jing-Zhou Chen, PhD

State Key Laboratory of Cardiovascular Disease,

Fuwai Hospital,

National Center for Cardiovascular Diseases,

Chinese Academy of Medical Sciences,
Peking Union Medical College,
Beijing 100037, China.
Telephone: 86-10-60866383
E-mail: chendragon1976@aliyun.com

References

1. Pyysalo, M. J. *et al.* Bacterial DNA findings in ruptured and unruptured intracranial aneurysms. *Acta Odontol. Scand.* **74**, 315-320 (2016).
2. Aboukais, R. *et al.* Absence of bacteria in intracranial aneurysms. *J. Neurosurg.* , 1-5 (2019).
3. Horvath, D. M., Murphy, R. M., Mollica, J. P., Hayes, A. & Goodman, C. A. The effect of taurine and β -alanine supplementation on taurine transporter protein and fatigue resistance in skeletal muscle from mdx mice. *Amino Acids* **48**, 2635-2645 (2016).
4. Jong, C. J., Ito, T., Mozaffari, M., Azuma, J. & Schaffer, S. Effect of beta-alanine treatment on mitochondrial taurine level and 5-taurinomethyluridine content. *J. Biomed. Sci.* **17 Suppl 1**, S25 (2010).
5. Shaffer, J. E. & Kocsis, J. J. Taurine mobilizing effects of beta alanine and other inhibitors of taurine transport. *Life Sci.* **28**, 2727-2736 (1981).
6. Knights, D. *et al.* Bayesian community-wide culture-independent microbial source tracking. *Nat. Methods* **8**, 761-763 (2011).
7. Staley, C. *et al.* Stable engraftment of human microbiota into mice with a single oral gavage following antibiotic conditioning. *Microbiome* **5**, 87 (2017).
8. Kaur, S., Yawar, M., Kumar, P. A. & Suresh, K. *Hungatella effluvii* gen. nov., sp. nov., an obligately anaerobic bacterium isolated from an effluent treatment plant, and reclassification of *Clostridium hathewayi* as *Hungatella hathewayi* gen. nov., comb. nov. *Int. J. Syst. Evol. Microbiol.* **64**, 710-718 (2014).
9. Crost, E. H. *et al.* Production of an antibacterial substance in the digestive tract involved in colonization-resistance against *Clostridium perfringens*. *Anaerobe* **16**, 597-603 (2010).
10. Sharon, G. *et al.* Human Gut Microbiota from Autism Spectrum Disorder Promote Behavioral Symptoms in Mice. *Cell* **177**, 1600-1618.e17 (2019).

Reviewers' Comments:

Reviewer #2:

Remarks to the Author:

Li et al. seem have addressed most of my comments sufficiently and have provided clear point-to-point rebuttal. However, I have some concerns about data integrity. I observed the exactly same network figure using SparCC (the sup Fig 4) as the previous network figure using Spearman at the initial submission (Fig 2). I wish this was just an accident. The inaccuracy was also spotted at the initial submission. The authors need to convince me that all data and results are correctly presented in the paper firstly.

Furthermore, several statements lack statistical evidence. For instance, UIA-enriched MLGs were more highly interconnected than control-enriched MLGs.

Reviewer #3:

Remarks to the Author:

The authors have been very responsive to my comments and those of the other reviewers. The manuscript is significantly improved. I have no further concerns.

Robert Jenq

Reviewer #4:

Remarks to the Author:

In the response to the reviewers' comments under point 5 that authors discuss down regulating taurine in mice to detect a difference in incidence of aneurysm formation, progression and rupture. The Figure accompanying point 5, panel c illustrates that the vehicle group incidence of no aneurysm, unruptured aneurysm and ruptured aneurysm totals to 105%. This should be corrected and probably included in the supplementary material. This further corroborates the critical role of taurine production through gut microbiota in cerebral aneurysm pathogenesis.

We particularly appreciate the constructive suggestions and insightful comments by the reviewers. The manuscript is now extensively revised and significantly strengthened, and we have addressed all concerns raised by the reviewer 2 and 4.

The following is the point-to-point response to the Reviewers' comments.

Reviewer #2 (Remarks to the Author):

Li et al. seem have addressed most of my comments sufficiently and have provided clear point-to-point rebuttal. However, I have some concerns about data integrity. I observed the exactly same network figure using SparCC (the sup Fig 4) as the previous network figure using Spearman at the initial submission (Fig 2). I wish this was just an accident. The inaccuracy was also spotted at the initial submission. The authors need to convince me that all data and results are correctly presented in the paper firstly.

Response: Thanks for pointing out this critical issue. We followed this reviewer's original comments and reconstructed the network structure using SparCC. But we did inadvertently use the previous network figure constructed by Spearman in R1. We must sincerely apologize for the misunderstanding and confusion caused by our negligence. Furthermore, we have submitted the correct version to the editorial board immediately after receiving the decision letter to eliminate the misunderstanding. We've corrected it in the revised manuscript (please see the revised Supplementary Figure 4).

Meanwhile, we have carefully double-checked the revised manuscript and we guarantee all data and results are correctly presented, and no violation of any publication ethics. We apologize to this reviewer again.

Furthermore, several statements lack statistical evidence. For instance, UIA-enriched MLGs were more highly interconnected than control-enriched MLGs.

Response: Thanks for pointing out this important issue. We have carefully checked the manuscript and corrected the statements and figures, which are summarized

below:

(1) Fig. 1a: We have deleted the statement “and was higher in the UIA group than in the control group” (please see page 6, line 2-3).

(2) Fig. 1f: We have labeled the accurate P value on Fig. 1f.

(3) Fig. 7b: We have labeled the accurate P value on Fig. 7b. Because the sample size of human donors were 2 in each group, we did not do statistical analysis between UIA patients and controls. Meanwhile, we have removed “UIA donors” from the statement “Furthermore, lower relative abundances of *H. hathewayi* all occurred in UIA patients in the two cohorts, UIA donors, and mice treated with UIA patient feces compared with their respective controls” (please see page 18, line 13-15).

(4) Supplementary Fig. 4: We have deleted the statement “Notably, UIA-enriched MLGs were more highly interconnected than control-enriched MLGs.” We have added the statement “We found 256 and 171 positive significant correlations in UIA and control-enriched MLGs, respectively (correlation coefficient > 0.5 and Benjamin-Hochberg corrected $P < 0.01$). Five negative significant correlations were also found among MLGs (correlation coefficient < -0.5 and Benjamin-Hochberg corrected $P < 0.01$)” (please see page 8, line 3-9).

(5) Supplementary Fig. 9: We have labeled the accurate P value on Supplementary Fig. 9.

Reviewer #3 (Remarks to the Author):

The authors have been very responsive to my comments and those of the other reviewers. The manuscript is significantly improved. I have no further concerns.

Robert Jenq

Reviewer #4 (Remarks to the Author):

In the response to the reviewers' comments under point 5 that authors discuss down

regulating taurine in mice to detect a difference in incidence of aneurysm formation, progression and rupture. The Figure accompanying point 5, panel c illustrates that the vehicle group incidence of no aneurysm, unruptured aneurysm and ruptured aneurysm totals to 105%. This should be corrected and probably included in the supplementary material. This further corroborates the critical role of taurine production through gut microbiota in cerebral aneurysm pathogenesis.

Response: We are sorry for this mistake and have corrected it (please see the revised Supplementary Figure 17c). Moreover, we fully agree with this reviewer's suggestion and have included these data to the main text (please see page 17, line 17-22 and page 18, line 1-4 and see the revised Supplementary Figure 17).

Reviewers' Comments:

Reviewer #2:

Remarks to the Author:

I am pleased by the responses and have no further comments.

REVIEWERS' COMMENTS:

Reviewer #2 (Remarks to the Author):

I am pleased by the responses and have no further comments.

Point by point responses to the Reviewers' comments:

Reviewer #2 (Remarks to the Author):

I am pleased by the responses and have no further comments.

Response: We appreciate that this reviewer is now satisfied with our revised manuscript.